# Feeling heard: Operationalizing a key concept for social relations

**Carla Anne Roos** [1] *, **Tom Postmes** [2], **Namkje Koudenburg** [2]

**1** Department of Communication and Cognition, School of Humanities and Digital Sciences, Tilburg University, Tilburg, North Brabant, The Netherlands, **2** Department of Social Psychology, Faculty of Behavioral and Social Sciences, University of Groningen, Groningen, The Netherlands

* c.a.roos@tilburguniversity.edu

## Abstract

Feeling heard is considered a cornerstone of close relationships and crucial to healthy self-development, but psychologically, this sentiment of feeling heard remains understudied. The current paper therefore aims to define and measure the experience of feeling heard. Based on an integrative literature review, feeling heard is conceptualized as consisting of five components at two conceptual levels. At the interpersonal level people feel heard when they have 1) voice, and receive 2) attention, 3) empathy, 4) respect. At the collective level people should experience 5) common ground. In two population surveys ($N$ = 194, $N$ = 1000), we find that feeling heard is a unitary concept, and we develop and validate the feeling-heard scale (FHS); a concise eight-item scale with good psychometric properties. Results show that the FHS is a distinct predictor of conversation intentions in many different contexts and relationships. In fact, the FHS is the strongest predictor of intentions for conflict behavior among a set of 15 related variables (e.g., acquaintance, intimacy). We conclude by reflecting on the potential applications of this scale: in interpersonal relations and professional contacts, the FHS enables the assessment of one crucial dimension of social interaction.

**Data Availability Statement:** All the materials, data and code used in this paper is publicly available in anonymized form on dataverse.nl: https://doi.org/10.34894/IHNKUN.

## Introduction

"*The right of all children to be heard and taken seriously constitutes one of the fundamental values of the Convention on the Rights of the Child*"

- Convention on the Rights of the Child, 2009 -

"*If we only listened with the same passion that we feel about being heard*"

- Harriet Lerner, n.d.; clinical psychologist -

"*In the final analysis a riot is the language of the unheard. And what is it that America has failed to hear*"

- Martin Luther King, Jr., 1968 -

**Funding:** Part of this research (i.e., Study 2) was funded by the Behavioral and Social Sciences Internet Research Fund of the University of Groningen (project code 170240169). The grant holder was Namkje Koudenburg; Tom Postmes and Carla Roos were co-applicants. The funder's website is https://www.rug.nl/gmw/?lang=en The funder had no role in study design, data collection and analysis, decision to publish, or preparation of the manuscript. No further grant was received from any funding agency in the public, commercial, or not-for-profit sectors.

**Competing interests:** The authors have declared that no competing interests exist.

References to feeling heard are made in many domains of everyday life. As the quotes above suggest, it is often seen as a cornerstone of intimate relationships, crucial to healthy self-development, and essential to a well-functioning representative democracy. Indeed, an abundance of self-help information and training exists to help people make others feel heard, for example romantic partners, children, employees, or customers, e.g., [1–4]. For those who want to feel heard themselves, many countries have so-called listening centers that offer a listening ear 24/7 (see https://www.ifotes.org/en). At a more abstract level, beyond interpersonal relationships and direct interactions, various individuals and groups feel not heard by their government or other institutions. Populist parties often gain popularity among these groups by purporting to represent those who are not heard by the elite, e.g., [5]. Alternatively, people that do not feel heard can unite and engage in collective action, e.g., [6]. For example, in the US, groups across the political spectrum, from Trump supporters to Black Lives Matter activists, protest to be heard, e.g., [7, 8].

Feeling heard thus appears to be a central concept in individualized Western societies. However, possibly due to a lack of systematic research into the concept (indeed, scientific study of the concept is scarce and scattered across different fields), we currently lack a good understanding of what the concept entails, and what its precursors and consequences may be. For example, in today's society many want to have a say and use social media platforms to express themselves to strangers on an unprecedented scale. But is all this self-expression on social media effective in making people feel heard? Recent research suggests that the opposite might be the case: a lack of instant feedback online can be misinterpreted as a signal of disinterest leading people to feel less heard compared to face-to-face conversation [9, 10]. Especially in the context of increasing political polarization and concerns about loneliness and isolation, e.g., [11], it would be important to measure the experience of feeling heard reliably.

In sum, feeling heard could be a key variable of our time and appears central to relationships, self-development, and representation. But psychologically, this sentiment of feeling heard remains understudied. This paper therefore embarks on exploring the concept, by integrating related constructs from different literatures (e.g., intimate relationships, healthcare, organizations) into a unified definition and measurement. We focus on the concrete level of feeling heard in everyday interpersonal interactions because feeling heard at an abstract level (e.g., by the government) might be different and therefore require an independent exploration.

## Feeling heard in the literature

Considering the importance of feeling heard in colloquial speech and common understanding, one would expect this concept to have its own scientific literature. But in a comprehensive review, we found that feeling heard is not treated as a distinct concept yet. Instead, the literatures on contexts where feeling heard could play an important role tend to revolve around related concepts. Exploring each of these literatures might help us define *feeling heard* and determine whether it is distinct from other existing concepts. In this section, we will shortly describe the key findings from our review.

**Intimate relationships literature: Perceived responsiveness.** First, the importance of feeling heard becomes evident from the literature on intimate relationships. Rather than focusing explicitly on feeling heard, this literature examines the closely related construct of perceived responsiveness. Perceived responsiveness has been defined as the belief that close others understand and value one's personal needs and goals and are supportive in fulfilling these [12]. It is essentially about a person's perception that intimate others see them for who they are and accept, value, and support that [13]. While perceived responsiveness was studied most in the context of romantic couples, it plays a role in all kinds of intimate relationships, such as

that between parent and child [14]. It has been found to support and strengthen not only the relationship, by increasing satisfaction, intimacy, and commitment, but also its members, by increasing subjective well-being and self-esteem, e.g., [15].

The concept of perceived responsiveness thus focuses on the target's perception of another's attitudes towards them. We believe this is also an important component of feeling heard in conversations. But feeling heard is more concrete than perceived responsiveness, in the sense that it is a feeling that arises from a specific interaction, rather than a general appreciation of a person's needs and goals. Recently, [16] suggested that the concrete form of responsiveness is *listening* and called for the integration of both concepts in future research. The perception of listening behavior might make a speaker feel heard. Listening has been extensively studied in the healthcare literature.

**Healthcare literature: Active listening.** Listening is mostly considered in service of the person-centered approach that is seen as a core dimension of high-quality healthcare [17]. The person-centered therapist is essentially a responsive therapist who puts the patient center stage and aims to stimulate the patient to pursue their own needs and goals [18]. This makes the client feel known and empowered [19]. The person-centered approach gained currency in many other domains beyond healthcare, for example, it is used in education to optimize child development and in retailing to increase customer satisfaction [20, 21].

The important practice in the person-centered approach is active listening (also called empathic listening or high-quality listening; [22]). This involves paying undivided, non-directive, and non-judgmental attention with behaviors like paraphrasing and summarizing what has been said [23]. Active listeners listen to the feelings and attitudes behind the speaker's words. The active listener tries to see the world through the speaker's eyes and empathize with them. In this way, the listener communicates respect for the speaker.

In this literature, the central focus is on listening behaviors or, more recently, on the perception of these listening behaviors (see for reviews [24, 25]). The experience of being listened to is treated as an important and positive but not yet clearly defined outcome, e.g., [26]. We suspect that to feel heard a speaker must *perceive* their interaction partner as an active listener. Much like perceived responsiveness then, to feel heard not only the behavior of the listener is important, but also the perception of the speaker.

Based on the previous, three ingredients seem required for one to feel actively listened to: attention, empathy, and respect. We expect these to be the first three components of feeling heard. First, the speaker will feel heard when they perceive (undivided) attention on the side of the listener. The listener is not a passive receiver but actively engaged; the listener is all ears. Second, the speaker will feel heard when they perceive the listener as empathic. The listener tries to step into the speaker's shoes and take their perspective. And third, the speaker must feel that the listener is respectful towards them. The listener takes the speaker seriously and does not judge them for what they say or who they are. Thus, the speaker feels heard when they perceive the listener as attentive, empathic, and respectful.

But to be listened to or heard, the speaker first needs to speak. This shifts the focus from the speaker's interpretations of the listener's behavior to the speaker themselves. Does the speaker feel able to express themselves?

**Organizational and law literatures: Voice.** The importance of being able to communicate per se is most explicitly recognized in the concept of *voice*, which is most relevant in contexts where there is a power difference and has therefore been examined in contexts of representation (e.g., in legal, political, and managerial settings). Voice is a central concept in the literature on procedural justice. Procedural justice is the fairness of the process by which a decision is made, and as such is contrasted with distributive justice which refers to the fairness

of the decision itself [27, 28]. Subjective procedural justice involves the *experience* of being treated fairly, which might be related to feeling heard. Voice contributes importantly to this experience [29]. To experience voice, subordinates should feel able to let their opinions and feelings be known and taken into account by those making decisions that affect them. Importantly, this does not require (the perception of) actually influencing the decision [30]. In organizations, experiencing voice tends to increase engagement, satisfaction, and social identification with the organization [31]. In the context of law, the importance of voice has also been recognized [32]. Without voice, litigants might feel treated as a number and not taken seriously, which can make procedures appear random and raise suspicions towards legal authorities.

Although voice is mostly studied in law or organizational contexts, the essence of being able to say what one wants to say could be an important component of feeling heard, that could apply to many everyday conversations. Integrating the three literatures described so far suggests that feeling heard involves being able to speak freely to an attentive other that shows empathy and respect. This all happens on an interpersonal level. However, to feel heard, interaction partners need a certain degree of understanding between them. Such understanding requires a superordinate level of the dyad or group: the collective.

**Communication literature: Common ground.** In communication science, this level of "us" is considered the basis for establishing a *common ground*. Common ground is the knowledge and beliefs that interaction partners assume to share, which allows them to communicate effectively [33, 34]. The continuous creation and maintenance of these mutual understandings, called grounding, occurs through the dynamics of interpersonal interaction. Grounding has been presented as a three-step process that starts with the speaker presenting information, followed by the listener expressing understanding or misunderstanding, which the speaker subsequently acknowledges or corrects [35]. Interaction is thus considered an act of coordination and collaboration [36].

It logically follows that to feel heard, interaction partners need to develop a common understanding of their needs and wishes in the conversation. This component seems to be quite distinct from the other, more transactional, foundations of feeling heard, which occur at the interpersonal level (i.e., the level of "me" and "you"). The communication literature adds a new element by pointing out that feeling heard may require a *joint effort* of grounding in which interaction partners dynamically construct a mutual understanding at the level of "us" (see also the literature on shared reality; [37]). We refer to this level of "us" or "we" as the "collective level" to distinguish it from interpersonal experiences. At the collective level interaction partners do not think about themselves and others as individuals but as one irreducible entity (also see the literature on collective intentionality, [38]).

At this point it is good to take stock: what have we learned so far? We believe that while none of these literatures are about *feeling heard* as such, all of them imply that it may be an important psychological outcome of social interaction and all of them add something important to understanding what feeling heard entails. One can feel heard because of particular *actions* of self (e.g., voice) and other (e.g., active listening). This exchange can feed into *perceptions* such as responsiveness. But feeling heard may also require an ongoing *dynamic* by which understandings of "us" are construed and maintained. In other words, in the psychological sense of feeling heard the actions and perceptions of self, other, and us jointly seem to converge. This conclusion implies two things: that *feeling heard* is not a redundant concept, because it is not reducible to any one of these components, and that operationalizing it should attend to each of these components. In addition to this literature review, we conducted semi-structured interviews with four individuals who had a lot of practical experience with and knowledge about feeling heard in different fields. The summarized interviews can be found in

the section 1.3 in S1 Appendix. The themes that came up in these expert interviews showed substantial overlap with the literatures outlined here.

## A definition of *feeling heard* and an operational model

Integrating the concepts found in these diverse literatures allows us to distinguish different components of *feeling heard* that should be included in a definition of this concept. *Voice* is a first vital part of feeling heard: one needs to communicate to be, and by extension feel, heard. Voice also presupposes a wish to be heard, a desire to communicate something. Indeed, the voice literature concerns situations where communication is not self-evident (because of status differences), not required (up to people themselves), but allowed and likely desirable (giving opinions on self-relevant decisions). So, voice is not just about piping up but about one's experience of having the opportunity to speak *if* one wants to speak, and to say *what* one wants to say.

There should also be someone receiving the communication. The literatures on intimate relations and healthcare elaborate on how receivers can put the speaker center stage. The receiver does so by paying close and undivided attention to the speaker and their communication. This is needed to accurately take the speaker's perspective and understand their needs. Lastly, the listener should be non-judgmental and accept the speaker for who they are. Whether the speaker feels heard is dependent on their perception and interpretation of the listener's behaviors. In sum, interpersonal perceptions of *attentiveness*, *empathy*, and *respect* are important components of feeling heard.

Finally, the communication literature suggests that feeling heard requires a sense that "we understand each other". This *common ground* appears related to all other components of feeling heard. It is essential to voice effectively: the speaker needs to adapt their communication to their comprehension of the listener's understanding to ensure the listener understands correctly. Common ground is also needed for attention, empathy, and respect: the listener must understand the speaker to attend to the correct details of their communication, empathize with them, and respect them for who they are.

Putting these different insights together, feeling heard has five components concerning three entities (me, you, and we). In order to feel heard, people take into account their interpersonal level experiences of *my* voice (1), and *your* attention (2), empathy (3) and respect (4). But beyond that, they may also factor in the collective level experience that *we* have (5) common ground. See Fig 1. This bring us to the following definition: feeling heard is the feeling that one's communication is received with attention, empathy, respect, and in a spirit of mutual understanding. In the remainder of this paper, we report on the development and validation of an instrument to measure this concept.

## Study outlines

The paper consists of two studies aimed at scale development and scale validation. We first constructed a 16-item scale based on the conceptualization of feeling heard described above. Study 1 was a Dutch population survey in which we tested the proposed model and scale to select the items and factor structure that best represent feeling heard. Study 2 was a preregistered US survey study designed to assess the reliability, convergent and divergent validities, and the predictive validity of the resulting feeling heard scale (the FHS). We additionally conducted a preregistered lab experiment in which we manipulated feeling heard by varying communication channel (audio vs. text) and tested whether feeling heard picks up on unique variance in conversational experiences that is not measured by other related variables. Because of space limitations, the description of this study and its results can be found in the sections 3.1 and 3.2 in S1 Appendix.

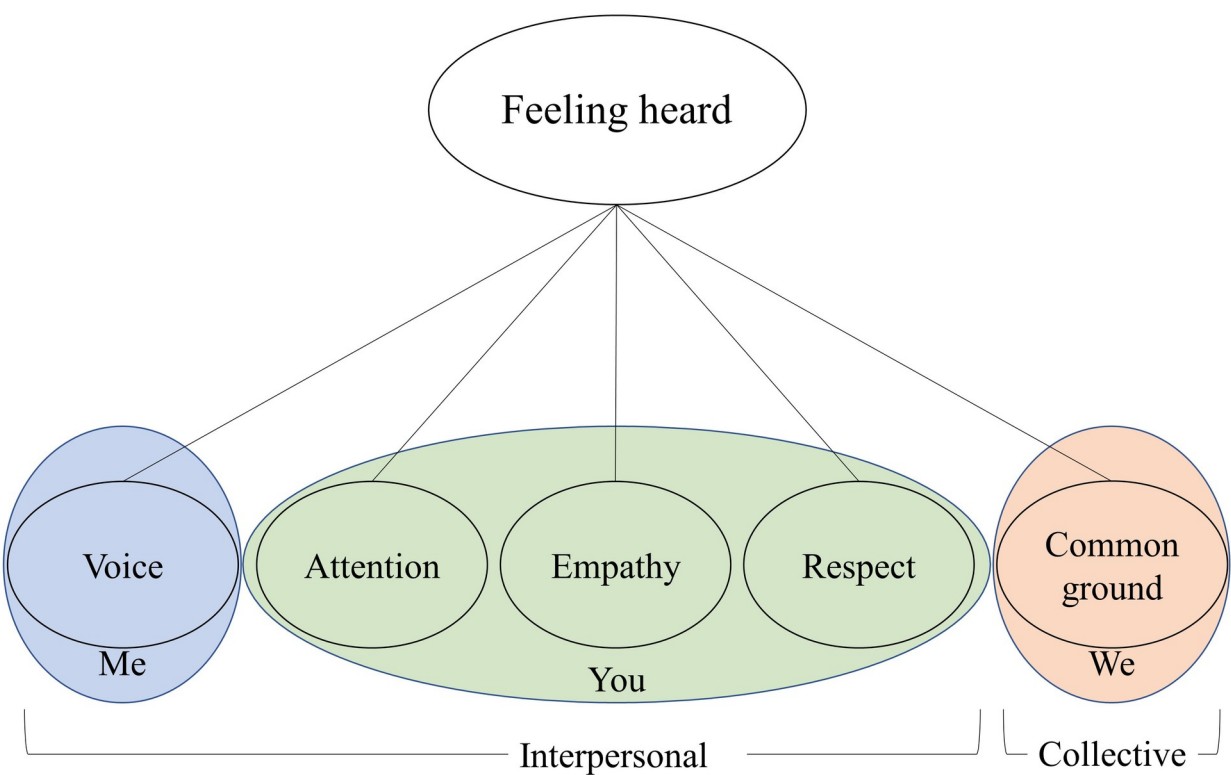

**Fig 1. Feeling heard in conversation.** A visual representation of the proposed model.

## Study 1

### Aims and hypotheses

Study 1 had two aims: 1) test the proposed factor structure, and 2) get an impression of the natural distribution of feeling heard in everyday online conversations. This first study was not preregistered. Because this study was conducted during the lockdowns due to the COVID-19 pandemic, we focused on *online* interactions, which then became a central mode of communication.

### Method

The studies reported in this paper and S1 Appendix were all approved by the Ethical Committee Psychology of the University of Groningen. The authors had no access to information that could identify individual participants during or after data collection. Informed consent was obtained from all participants prior to their participation. The materials, analysis code, data, and codebook for interpreting the data of all studies is publicly available on Dataverse at https://doi.org/10.34894/IHNKUN.

**Sample size rationale.** As a crude rule of thumb, for scale development at least 10 participants per item are required, which means 160 participants for 16 items [39]. We decided to recruit slightly over 200 participants, anticipating some attrition due to failed attention checks. With a sample of 160, correlations of $r = |.19|$ (small effect; [40]) can still be reliably estimated with alpha = .05 (two-tailed) and power = .80.

**Participants.** A representative sample of the Dutch population in terms of age, gender, and educational level was recruited and paid through the online panel service PanelInzicht (*N* = 217). Participant recruitment and data collection took place in May and June 2020. Participants were required to be native Dutch and to have at least some experience with online conversations. Based on their descriptions of a past online conversation, we excluded 23 participants who provided nonsensical answers or indicated that they did not have the required experience (e.g., "Sorry, no example, never happened" P.198). The final sample consisted of 194 participants, 51.55% female and 48.45% male. 37.11% were between 18 and 38 years old, 43.81% were between 39 and 59, and 19.07% were 60 or older. Most participants (46.91%) had a middle education level, 29.90% were higher educated, and 23.20% were lower educated (classification based on [41]). Most participants (73.20%) had a full- or part-time job, 18.04% were either unemployed or retired, and 8.76% were studying.

**Design and procedure.** The survey consisted of two parts. In Part A participants were either asked to think back to an online conversation in which they felt heard (*N* = 108) or one in which they did not feel heard (*N* = 86). This division was based on random allocation and done to maximize the variance in feeling heard experiences and to test the differences between (feeling heard and feeling not heard) conditions. In Part B we examined the natural distribution of feeling heard in online conversations by asking all participants to think back to their last online conversation. All participants completed both Part A and Part B but in randomized order. Eighty-nine participants began with Part A and 105 participants began with Part B.

The study was conducted in Dutch. Participants completed an online survey via the platform Qualtrics. After providing their demographics, participants, in both Part A and B, wrote down details about the conversation situation. In Part A, participants were additionally asked to write down *why* they felt heard or not. We performed an exploratory thematic analysis on the answers to these open-ended questions in both Part A and B of the dataset. In both Part A and B, participants subsequently answered some general questions about the described conversation (see Conversation characteristics below), and then completed the feeling heard scale. The entire questionnaire can be found in the section 1.2 in S1 Appendix. The questionnaire contained six additional measures to assess the impact of the COVID-19 pandemic on our sample. This data is not reported here because of lacking relevance.

**Materials.** *Conversation characteristics.* We included the following variables that might affect the experience of feeling heard: 1) quality of the internet-connection, 2) number of interaction partner(s), 3) acquaintance with the interaction partner(s), and perceived 4) quality and 5) equality of the relationship with the interaction partner(s). These variables also provide a characterization of the types of conversations described, see S1 Table in S1 Appendix (Table and Figure numbers preceded by an S can be found in the S1 Appendix).

Feeling heard. Participants were first presented with the central item that formed the kernel of our scale "In this conversation, I felt heard by the other(s)" (item 1). The rest of the feeling heard scale consisted of the five components described before. For each component, we designed three items. We also included four negatively phrased items. The items were phrased according to the entity their component represented: a) voice items referred to "I", b) attention, empathy, and respect items were about "the other(s)", and c) common ground items assessed "we".

All items were preceded by the phrase "In this conversation...". Depending on the number of interaction partners in the conversation, items referring to the interaction partner(s) were phrased with "the other" or "the others". All items were measured on 5-point Likert scales (from 1 = *Completely disagree* to 5 = *Completely agree*). See Table 1 for the operationalizations and items of the components.

**Table 1. The operationalization and original items of the five components of feeling heard.** The item numbering will be used to refer to the specific items in this results section. The last column contains the item numbers that were part of the final scale.

| Component | Operationalization | Initial items | Item number | Items in the final scale |
|---|---|---|---|---|
| Voice | My experience of being able to express myself freely, that is, being able to say what I want to say. | . . .I could say what I really wanted to say | 2 | 2 |
| | | . . .I could express my thoughts | 3 | |
| | | *. . .I felt inhibited to say what I wanted to say* | 4 | |
| Attention | My impression that the other focused their attention on what I said (my voice). | *. . .the other was more concerned with him/herself than with what I said* | 5 | 5, 6 |
| | | . . .the other listened to what I said | 6 | |
| | | . . .the other paid attention to what I said | 7 | |
| Empathy | My perception that the other tried to take my perspective and emotionally understand me. | . . .the other tried to put him/herself in my shoes | 8 | 8, 9 |
| | | *. . .the other was insensitive to my thoughts and feelings* | 9 | |
| | | . . .the other was empathetic | 10 | |
| Respect | My feeling that the other valued what I said (my voice) and me as a person. In other words, I am worth listening to. | . . .the other showed genuine interest in me | 11 | 13 |
| | | . . .the other took me seriously | 12 | |
| | | . . .the other treated me with respect | 13 | |
| Common ground | My impression that we could take each other's perspective and understand each other's point of view | *. . .we looked at things differently* | 14 | 15 |
| | | . . .we understood each other | 15 | |
| | | . . .we were on the same wavelength | 16 | |

*Note.* Reverse coded items are italicized. We show the items in singular form (i.e., "the other), some of the items were rephrased to plural form (i.e., "the others") when participants described a conversation with more than one other person.

## Results

We first describe the results of the exploratory thematic analysis on the answers to the open-ended questions in both Part A and Part B of the dataset. Then we present the results of the factor analyses performed on Part A of the data to select the items and factor structure that best fit the data and best represent the concept. Lastly, we used Part B of the data to explore the reliability of the scale and the natural prevalence of feeling heard in online conversations.

**Thematic analyses.** First, we analyzed the descriptions of the conversation situation by coding a) the type of relationship and b) the context of conversation (private or business). See Table 2 for the themes, their respective descriptions, and some examples. Second, we analyzed the reasons for feeling (not) heard through an inductive process, by iteratively coding and clustering answers. We allowed different themes to emerge per condition. Interestingly, these themes turned out to fall into the same four overarching categories.

All descriptions were independently coded by the first author and a second (also native Dutch) coder. Themes were coded as 0 if absent and 1 when present in the description. The themes were non-exclusive, meaning that multiple themes could be present in one description, which often occurred. All Kappas were satisfactory (see Tables 3 and 4). For the analysis, we used the average of both coders, such that a description would get a score of .50 when only one of the coders coded it as present.

Answers that both coders considered too vague or abstract to code were excluded from the coding dataset. This resulted in differing sample sizes for conversation versus reasons descriptions. Note that uncodeable cases were still valid for the quantitative analyses, as nonsensical answers had already been removed in a previous step. When only one of the coders deemed an answer uncodeable, they discussed until agreement.

**Table 2. The themes used to code the conversation descriptions on relationship type and general context.**

| Theme | Description | Examples |
|---|---|---|
| *Relationship type* | | |
| Service provider(s)—one-off | One-off exchanges with strangers, although another interaction might happen in the short term about the same issue. Oftentimes the participant is in a position of dependence. | A helpdesk of a (web-) shop, an advertiser, the police, a bank |
| Service provider(s)—repeated | Contacts are expected to be repeated over time and over issues. Participants feel a sense of connection with this person or the organization they represent. Oftentimes the participant is in a position of dependence. | A representative of the housing association, the bank, the neighborhood police, the municipality, a regular customer |
| Colleague(s) | Contacts are in-group members of equal or subordinate status to the participant. | Colleagues, classmates, fellow organizers |
| Supervisor(s) | Contacts are in-group members to whom the participant was subordinate. | Boss, teacher |
| Friend(s) | Contacts are in-group members with whom the participant is interdependent. | Friends |
| Family | Contacts are in-group members with whom the participant is interdependent. | Children, parents, cousins, spouse |
| Rest | Relationships that were only mentioned by a single or a few participants. | Sports coach, online community, date, researcher |
| *General context* | | |
| Private | A relatively "closed" or personal conversational context and/or topic. | Asking a doctor for medical advice |
| Business | A more open conversational context and/or topic than *Private*. | Work meeting with colleagues |

**Conversation situation.** Table 3 shows the percentages of descriptions referring to the themes per condition. Three results stand out. First, considering these are quite rare contacts, as evidenced in their last conversation frequencies, the service provider(s)—one-off was rather frequently mentioned when people thought back to a situation in which they either felt heard or not. This suggests that interactions with strangers where one is in a dependent role, relatively easily elicit experiences related to feeling heard. A similar discrepancy is visible for the supervisor(s) where one again is in a dependent, or subordinate, position. Second, the results show that people thought relatively less of conversations with intimate others (i.e., family and friends) when asked about a conversation in which they did not feel heard, compared to what would be expected based on the last conversation frequencies. In contrast, conversations in which people did not feel heard were relatively frequently business-focused. Third, all conditions most often featured colleagues. Indeed, over half of the remembered Last conversations were with colleagues and family. Thus, presumably due to the COVID-19 lockdown, many online conversations involved work meetings with colleagues.

**Reasons for feeling (not) heard.** We identified four overarching reasons for feeling heard or not heard, but as the sub-themes differed slightly between these conditions, the content of the themes also differs. Below we list the four themes and shortly describe their respective sub-themes, indicating differences between conditions. We also provide some illustrative quotes. Table 4 shows the occurrence of the themes per condition. The reasons descriptions provided by participants that did not yet see our feeling heard items (first measurement) did not differ significantly from the reasons provided by participants that did (repeated measurement) and whose conception of feeling heard could have been influenced by our items.

*1. (Lack of) respect.* As reasons for both feeling heard and not heard, participants mentioned respectful treatment. Participants wrote about being taken seriously and receiving recognition. They also mentioned patience on the side of interaction partner(s).

"*I had the impression the lady on the phone thought: 'O this is another one with a problem due to the current circumstances.' While I believe that every person must be helped, people simply need support in this period. [. . .]*"

Not feeling heard, P. 167

**Table 3. The interrater reliabilities and percentages of occurrence of each relationship and context theme in each conversation description: Their last conversation, a conversation in which they felt heard, and a conversation in which they felt not heard.**

| Code | Kappa [95%CI] | Last ($N = 206$) | Felt heard ($N = 103$) | Felt not heard ($N = 80$) |
|---|---|---|---|---|
| *Relationship type* | | | | |
| Service provider(s)—one-off | .81 [.72, .90] | 6.80%[a] | 12.62%[a] | 12.50%[a] |
| Service provider(s)—repeated | .82 [.74, .90] | 13.59%[a] | 11.65%[a] | 17.50%[a] |
| Colleague(s) | .92 [.88, .96] | 32.04%[a] | 24.27%[a] | 31.25%[a] |
| Supervisor(s) | .84 [.75, .93] | 5.83%[a] | 9.71%[a, b] | 13.75%[b] |
| Friend(s) | .87 [.79, .96] | 7.77%[a] | 10.68%[a] | 5.00%[a] |
| Family | .95 [.91, .99] | 20.39%[a] | 14.56%[a] | 5.00%[b] |
| Rest | .81 [.71, .91] | 6.31%[a] | 8.74%[a] | 12.50%[a] |
| *General context* | | | | |
| Business | .81 [.75, .87] | 50.97%[a] | 55.34%[a, b] | 67.50%[b] |
| Private | .83 [.77, .88] | 41.75%[a] | 37.86%[a, b] | 28.75%[b] |

*Note.* The uncodeable conversation descriptions were excluded. Percentages do not add up as the categories were non-exclusive. The percentages are fractions of the total number of described conversations in each category, so 206 for Last, 103 for Felt heard and 80 for Felt not heard conversations. Percentages in the same row that do not share the same superscript significantly differ at $p < .05$, based on a 2-sample z-test for equality of proportions.

"*By arriving late and not having enough time for me, I got the impression that I am not important.*"

Not feeling heard, P. 184

For feeling heard specifically, participants mentioned a more general friendly or polite treatment. Some also mentioned clear communication, as well as a sincere and honest interaction.

A sub-theme that was found almost exclusively in the not feeling heard condition was that of dominant interaction partner(s). This could either take the form of the partner(s) being too self-involved to take notice of the participant or being pushy or intrusive in trying to influence the participant.

*2. (Lack of) engagement.* Active engagement of interaction partner(s) was evidenced by reactions, asking follow-up questions, remembering, or summarizing what was said, and other signs of attention. Interestingly, (lack of) active engagement featured less in the reasons for not feeling heard (Table 4).

**Table 4. The interrater reliability and the percentage of occurrence of the reasons' themes per condition.**

| Code | Feeling heard ($N = 101$) | | Feeling not heard ($N = 88$) | |
|---|---|---|---|---|
| | Kappa [95%CI] | % occurrence | Kappa [95%CI] | % occurrence |
| 1. (Lack of) Respect | .63 [.45, .80] | 27.23%[a] | .66 [.50, .82] | 36.36%[a] |
| 2a. (Lack of) Engagement *Active* | .88 [.79, .97] | 45.54%[a] | .85 [.72, .99] | 19.32%[b] |
| 2b. (Lack of) Engagement *Passive* | .86 [.76, .96] | 50.00%[a] | .86 [.76, .97] | 48.86%[a] |
| 3. (Lack of) Effort | .76 [.64, .89] | 48.51%[a] | .59 [.40, .79] | 23.30%[b] |
| 4. (Lack of) Common ground | .87 [.78, .97] | 37.62%[a] | .82 [.69, .94] | 31.25%[a] |

*Note.* The uncodeable reasons descriptions were excluded. Percentages do not add up as the categories were non-exclusive. Percentages in the same row that do not share the same superscript significantly differ at $p < .05$, based on a 2-sample z-test for equality of proportions.

"*He reacted to what I said and he asked follow-up questions.*"

Feeling heard, P. 157

There was also a more passive form of engagement. This involved interaction partner(s) giving the participant room to speak, by listening, not interrupting, and letting them finish.

"*During the conversation, I could tell everything I wanted to tell. I had plenty of time for this and I didn't have to rush my conversation. [. . .]*".

Feeling heard, P. 27

A frequent reason for feeling not heard was the presence of too many people and no chair in online meetings, sometimes in combination with a bad internet connection. In such a conversation people talk all at once, meaning no one is heard. Relatedly, participants felt not heard when others were distracted or paid attention to something or someone else.

"*Everyone talked at once, was working on something else, people who were eating. Everyone was just talking. It was not pleasant. People were busy with everything, except with what they should be doing.*"

Not feeling heard, P. 197

*3. (Lack of) effort*. Quite often, participants described a conversation in which they wanted to get something done. As such, participants felt heard when their interaction partner(s) came to action or promised to do so.

"*Time and again I was sent from pillar to post and previously made agreements were not kept.*"

Not feeling heard, P. 91

In the feeling heard condition, participants also mentioned interaction partner(s) initiating help without request, such as comforting or giving tips.

Participants felt heard when they attained their predefined goal, or when they received a satisfactory answer for why they did not get what they wanted. Getting good quality answers could also be a goal in itself.

"*There was some sort of standard answer. In the sense that they could no longer reach the supplier. And so there was nothing more that could be done.*"

Not feeling heard, P. 173

*4. (Lack of) common ground*. Interaction partners could create a sense of common ground by recognizing the participant's situation or taking the participant's perspective. In the feeling heard context, understanding also occurred on an emotional level (i.e., empathizing).

"*This girlfriend asked if she could do something for me because I seemed so grumpy about the quiz team.*"

Feeling heard, P. 139

Some participants referred to the level of agreement between themselves and their interaction partner(s). Conversely, "closed" and stubborn others who stood firm to their own opinion would make the participant feel not heard.

"*I did not agree with supervisor (or supervisor not with me). I was supported by a few colleagues, but supervisor stood firm. After 2 protests I stopped trying.*"

Not feeling heard, P. 174

Moreover, participants in the feeling heard condition explicitly mentioned that they felt heard when their interaction partner(s) collaborated with them, for example by making decisions or appointments together or by negotiating.

"*To come to a decision together, the other has to listen and when you actually come to a decision together, that means that the other has heard you.*"

Feeling heard, P. 134

In sum, the themes and sub-themes that emerged from this content coding of the reasons' descriptions overlap considerably with our definition (and scale) of feeling heard, except for empathy, which is not a distinct theme but more a special case of understanding. Moreover, the effort theme is not present in our scale. Central to this latter theme is the accomplishment of goals, that is, getting out of the conversation what one wanted or expected.

**Scale dimensionality and item selection.** *Item characteristics.* See S2 Table in S1 Appendix for the item means, standard deviations, skewness, kurtosis, and between-condition differences. Condition (feeling heard vs. not feeling heard) had a significant effect on all items (ranging from $t(124.84) = |23.12|$, $p < .001$ to $t(165.83) = |6.17|$, $p < .001$). This suggests that all items tapped into the experience of feeling heard as they were successful in distinguishing between feeling heard versus non-heard experiences. Both the Bartlett's Test of Sphericity ($\chi^2$ $(120) = 3431.06$, $p < .001$) and the Kaiser-Meyer-Olkin test (MSA ranged between .84 and .98) showed that there was enough overlap between the items for conducting factor analysis [42]. This is also visible in the inter-item correlations that were all significant with a mean of $|.63|$ (see S3 Table in S1 Appendix).

**Factor analysis.** We conducted a confirmatory factor analysis with the Lavaan package in R (version 0.6–9; [43]). Because the scale was build upon our theoretical model for feeling heard, we used confirmatory factor analysis rather than exploratory factor analysis. To correct for the lack of multivariate normality in our data we used maximum likelihood estimation with robust standard errors and a Satorra-Bentler scaled test statistic (MLM) for the model parameters.

Alongside the chi-square test ($\chi^2$), we report the Comparative Fit Index (CFI) because this alternative fit index is less affected by the reliability paradox [44]. This is the phenomenon that more reliable measures lead to worse fit: higher standardized factor loadings provide more reliable information about a latent variable and therefore a higher power to detect small misspecifications in the model. Especially the absolute fit indices, like RMSEA, are sensitive to this paradox [45]. We will additionally inspect modification indices and residual correlation matrices to assess fit.

**Hypothesized model fit.** We tested the fit of the hypothesized model with a confirmatory factor analysis. Corresponding to the five feeling heard components, we specified five latent factors: Voice, Attention, Empathy, Respect, and Common ground, each with three unique items. All latent factors were additionally set to load on item 1. As can be seen in Fig 2, the fit

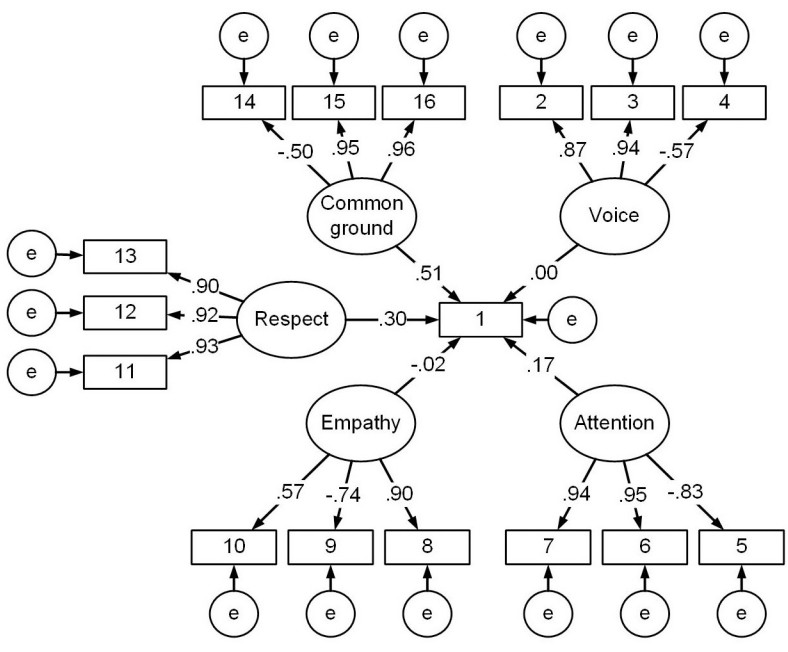

**Model fit indices**
$X2(90) = 195.70$, $p < .001$
CFI = .96

Covariances latent factors

| | |
|---|---|
| Voice – Attention: | .81 |
| Voice – Empathy: | .75 |
| Voice – Respect: | .71 |
| Voice – Common ground: | .69 |
| Attention – Empathy: | .94 |
| Attention – Respect: | .92 |
| Attention – Common ground: | .85 |
| Empathy – Respect: | .99 |
| Empathy – Common ground: | .94 |
| Respect – Common ground: | .91 |

**Fig 2. The results of the confirmatory factor analysis of the hypothesized model.** *Note*. Rectangles represent observed variables and circles represent latent variables. The loadings of items 2, 6, 19, 13, and 15 were set to 1, as these were considered the defining items of their respective components. Estimation methods: Maximum likelihood with robust standard errors and a Satorra-Bentler scaled test statistic (MLM). Both latent and observed variables are standardized.

of this hypothesized model was acceptable. However, the covariances among the five latent factors were notably high (ranging from .69 to .99), which might be indicative of unidimensionality. The modification indices and residual correlations (S4 and S5 Tables in S1 Appendix) also suggested considerable overlap between and within all latent factors.

Moreover, as can be seen in Fig 2, the factor loadings of item 1 were unreliable. Because this item was set to load on all latent factors and all latent factors were very strongly related, the item's explained variance is distributed over all latent factors. It is a zero-sum situation: the variance explained by one factor overlaps with that explained by another, but the variance can only be explained once.

**One-factor model fit.** Since the results above suggested item redundancy, we inspected the inter-item correlations (see S3 Table in S1 Appendix). Correlations of $r \geq |.80|$ (equaling a shared variance of 64%) were considered indicative of redundancy. Correlations of this magnitude that involved item 1 were not considered problematic as this is the scale's central item. If the other highly correlated item pairs were also conceptually considered redundant, we excluded one of the items. This led to the exclusion of items 3, 7, 11, 12, and 16. There were also items with consistently low inter-item correlations of $r \leq |.40|$ (equaling a unique variance of 84%): items 4, 10, and 14. These were excluded too.

The number of factors to extract from the remaining eight items was based on a scree-plot and Kaiser's criterion to remove components with eigenvalues $< 1$ (using the nFactors package in R, version 2.4.1, [46]). This suggested a one-component solution (see S1 Fig in S1 Appendix), which is in line with the interpretations above. The eight items were accordingly submitted to a confirmatory factor analysis with a one-factor solution. The fit of this model was satisfactory ($\chi^2(20) = 46.72$, $p < .001$; CFI = .98), but the modification indices and the residual correlations suggested a theoretically sensible correlation between the residuals of the two reverse coded items (see S6 and S7 Tables in S1 Appendix). Allowing this correlation resulted in a significantly better model fit ($\chi^2(1) = 5.76$, $p = .02$). The fit was satisfactory on all indices: $\chi^2(19) = 34.53$, $p = .02$; CFI = .99; all modification indices $< 15$; all residual correlations $< |.09|$, see Fig 3, and S8 and S9 Tables in S1 Appendix. The resulting feeling heard scale

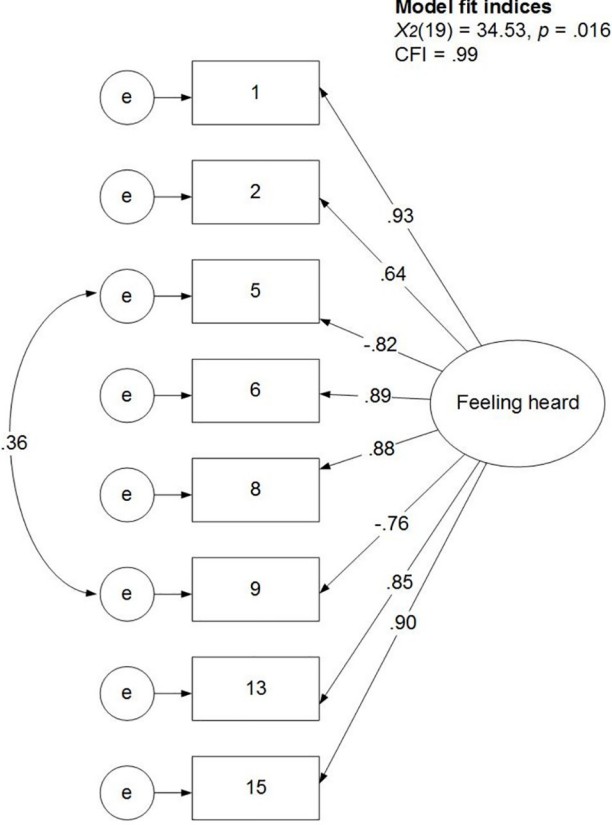

**Model fit indices**
$X2(19) = 34.53$, $p = .016$
CFI = .99

**Fig 3. Results of the confirmatory factor analysis of the final model.** *Note.* Rectangles represent observed variables and circles represent latent variables. Estimation methods: Maximum likelihood with robust standard errors and a Satorra-Bentler scaled test statistic (MLM). Reported are the completely standardized solutions: both latent and observed variables are standardized.

was highly internally consistent with an average item-total correlation of .72 (and a range of .53 to .78), and a coefficient omega of .95, 95%CI [.93, .96]. Factor loadings ranged from .64 to .93 with the highest loading for item 1 which was also conceptually the central item of the scale. This item, with an item-total correlation of .75, explained 56% of the variance in feeling heard as measured by the full scale.

**Influence of conversation characteristics.** We explored the correlations between feeling heard and other characteristics of the conversation or the social relationship. Participants were slightly more likely to feel heard when they had a good internet connection ($r = .15$, $p = .04$). Feeling heard was not related to the number of interaction partners ($r = -.07$, $p = .30$), nor to whether participants knew these partners before their conversation ($r = .08$, $p = .25$). Participants felt more heard when they had a good relationship with their interaction partner(s) ($r = .38$, $p < .001$). Lastly, participants also felt more heard when they felt this relationship was equal than when they considered either their interaction partner(s) or themselves of greater authority ($F(3,190) = 5.74$, $p < .001$; $M_{equal} = 3.70$, 95%CI [3.51, 3.88] vs. $M_{other-authority} = 3.26$, 95%CI [2.96, 3.56], Cohen's $d = .44$, small effect, and $M_{self-authority} = 3.30$, 95%CI [2.59, 4.01], Cohen's $d = .39$, small effect). Participants that indicated "don't know" to this question of equality felt least heard ($M_{don't-know} = 2.78$, 95%CI [2.33, 3.23]). The demographic variables (age, gender, education, and employment status) did not correlate with feeling heard. In sum, having a good internet connection, and a positive and equal relationship with one's interaction partner(s), seems to make it more likely that one feels heard.

**Last conversation description.** We analyzed Part B of the dataset to explore the reliability of the scale and the natural prevalence of feeling heard in online conversations. Excluding the nonsense descriptions of last online conversations (3 out of 217 observations) resulted in a sample size of 214 participants for Part B. The single-factor model fitted the data well and the loadings were very similar to Part A. Part B also allowed a tentative indication of the measurement invariance of the scale or "the psychometric equivalence of a construct across groups or across time" ([47], p.1). We used the measurementinvariance function of the semTools package (version 0.5–5, [48]). Results show that the model specified on Part A of the dataset also provides a good fit to Part B. This suggests that the scale's psychometric characteristics replicate across measures within the same sample, see S10 Table in S1 Appendix. The scale was internally reliable with an omega of .91, 95%CI [.88, .94]. The item-total correlation averaged |.62| and ranged from |.46| to |.71|. See S11 Table in S1 Appendix for the item means, standard deviations, and correlations. The distribution of feeling heard in Part B was negatively skewed (skewness = -1.10; $M = 3.92$, $SD = 0.82$) and the mean level of feeling heard was significantly higher than the scale midpoint ($t(213) = 16.37$, $p < .001$, Cohen's $d = 1.12$, large effect). This suggests that people tend to feel heard in their online conversations.

## Discussion

People do not seem to meaningfully distinguish between the five components—voice, attention, empathy, respect, common ground—deduced from the literature review. This is in line with recent studies showing that people experience listening holistically [49]. Feeling heard appears to be a unitary concept that can be measured with a concise eight-item scale. This scale does, however, contain elements of all five components and therefore taps into three entities (me, you, and we) at two conceptual levels (interpersonal and collective). These findings corroborate our definition of feeling heard as stated in the introduction. Providing initial evidence of the reliability of the scale, a good fit to Part B of the dataset was obtained. This is of course imperfect evidence since the same sample was involved. Study 2 therefore examines whether the scale is also reliable in another, unrelated dataset.

We additionally performed a qualitative analysis of participants' descriptions of feeling heard. Although this combination of model testing (quantitative data) and generation (qualitative data) in a single study can be seen as a limitation, the derived themes overlapped to a great extent with our definition. One new theme came up: getting out of the conversation what one wanted or expected. In light of procedural justice being as important as distributive justice [30], we do not expect goal accomplishment to be a required component of feeling heard. That is, people can feel heard when they do not accomplish their goals (no distributive justice), and, vice versa, people can feel not heard even though they do get what they asked for (distributive justice) because of the way they are treated (procedural justice). To verify that goal accomplishment and feeling heard are not the same, we include a goal accomplishment measure as part of the convergent validity testing in Study 2.

## Study 2

### Aims and hypotheses

Study 2 aimed to establish the 1) reliability, 2) convergent and divergent validity, and 3) predictive validity of the feeling heard scale. Unlike Study 1, the context was not restricted to a certain medium but concerned conversation in general. The design and hypotheses of this study were preregistered prior to data collection at the Open Science Framework: https://doi.org/10.17605/OSF.IO/N4R32. We formulated three hypotheses:

Hypothesis 1: The eight items load on a single latent factor and the feeling heard scale has good reliability.

Hypothesis 2A: The feeling heard scale will moderately to strongly and positively correlate with, but not be identical to: A) perceived intimacy of interaction partner(s), B) attraction towards interaction partner(s), C) perceived responsiveness of interaction partner(s), D) experienced goal accomplishment in conversation.

Hypothesis 2B: The feeling heard scale will moderately to strongly and negatively correlate with, but not be identical to: E) perceived dominance of interaction partner(s), F) distrust towards interaction partner(s), G) communication apprehension.

Hypothesis 3: The less heard participants feel in a conversation, the stronger their intentions to *negatively* avoid having another conversation with the same interaction partner(s). Negative avoidance means avoidance without relational maintenance efforts, i.e., without being polite or friendly.

To establish its discriminant validity, we additionally tested to which conversational variables (e.g., number of interaction partners) feeling heard does *not* relate. This was an exploratory analysis and not included in the preregistration.

### Method

**Sample size rationale.** As we wanted to sample the different types of conversations in which people felt heard and not heard, we decided to recruit a sample of 1000 respondents. This enables us to compare the most often mentioned types with a decent *N* in potential future analysis. With this sample size, correlations of |.08| (small effect; [40]) can be reliably estimated with alpha = .05 (two-tailed) and power = .80. Because of this high power, we interpret the effects in terms of their effect size rather than their statistical significance alone.

**Participants.** Participants were recruited via the online platform Prolific in December 2020. As the questionnaire was in English, the selection criteria were being fluent in English

and living in the United States. We collected 50% female and 50% male participants. We excluded 28 participants that did not provide serious answers (as visible in missing conversation descriptions and/or two failed attention checks) and replaced them with new participants. The final sample therefore consisted of 1000 participants ($M_{\mathrm{age}}$ = 34.89, $SD_{\mathrm{age}}$ = 12.94) of which 49.30% were female and 49.10% were male (1.60% indicated either "other" or "do not want to say"). Most participants (59.00%) were higher educated (undergraduate, graduate, or doctoral degree obtained), 21.40% were of middle educational level (college degree), and 19.60% were lower educated (no formal qualifications or secondary school completed) (classification based on [50]). About two-thirds of the sample (66.50%) was either full- or part-time employed, 21.50% was unemployed or retired, and 12.00% was studying.

**Design and procedure.** Participants completed an online survey via the Qualtrics platform. Participants read the following instruction: *Think back to a conversation in which you wanted to share your thoughts or feelings, make a point, speak your mind, and/or get something done (regardless of whether this was successful or not). Describe this conversation in a couple of sentences below.* This instruction was designed to make people think back to conversations in which feeling heard is relevant, and to exclude things like a hello-goodbye "conversation" with the cashier. With this conversation in mind participants then completed the rest of the questionnaire, described below. Lastly, participants provided their demographics and were thanked, debriefed, and paid. The entire questionnaire can be found in the section 2.2 in S1 Appendix.

**Materials.** *Reliability*. Participants were first presented with the eight-item feeling heard scale, translated to English (1 = *Strongly disagree* to 5 = *Strongly agree*). Throughout the questionnaire, items about interaction partner(s) were phrased in singular ("the other person") or plural ("the other people") form, depending on whether the described conversation was held with one or more than one person respectively. The example items presented here are all phrased in singular form.

*Convergent and divergent validity*. After careful consideration, we decided to include the following seven scales to assess convergent and divergent validity (see text 2.3 in the S1 Appendix for our inclusion criteria and reasoning). As indicator of scale reliability, we report omega (hierarchical) with bias corrected and accelerated bootstraps (1000), calculated with the ci.reliability function of the MBESS package (version 4.8.0, [51]).

First, two subscales of the relational communication scale of [52]: 1) conversational intimacy (20 items, 1 = *Strongly disagree* to 7 = *Strongly agree*), e.g., "The other person was intensely involved in our conversation" (ω = .95), and 2) dominance vs. equality (nine items), e.g.,: "The other person tried to control the interaction" (ω = .69). Second, the individualized trust scale of [53] (15 items, 7-point scales without anchors; [54]). This is a semantic differential scale asking to what extent each characteristic applies to a certain person, for example: "Trustworthy-Untrustworthy" (ω = .97). Higher scores indicate a more distrustful attitude. Third, the affective attraction scale of [55] (five items, 1 = *Extremely [unpleasant]* to 9 = *Extremely [pleasant]*), for example: "How unpleasant/pleasant do you feel about the other person?" (ω = .89). Fourth, the perceived partner responsiveness scale of [13] (18 items, 1 = *Not at all true* to 9 = *Completely true*), e.g., "The other person is responsive to my needs" (ω = .98). Fifth, a self-devised perceived goal accomplishment measure (three items, 1 = *Strongly disagree* to 5 = *Strongly agree*): "In this conversation, I accomplished the goals I had prior to this conversation", "I was unsuccessful in achieving what I wanted in this conversation" (reverse coded), and "I got what I aimed for in this conversation" (ω = .90). Last, the dyad subscale of the communication apprehension scale by [56] (six items, 1 = *Strongly disagree* to 5 = *Strongly agree*), for example, "Ordinarily I am very tense and nervous in conversations" (ω = .93).

*Predictive validity*. One of the most important consequences of not feeling heard is that it may catalyze conflict. Specifically, we expect that people who feel not heard are likely to engage in interaction behaviors that may contribute to conflict. To test this, we developed items to assess behavioral intentions for a next conversation with the same partner(s).

The items were based on the classic distinction between approach and avoidance reactions. Within these reactions, we distinguished between *positive* approach/avoidance: behaviors that steer towards maintenance of the relationship between the participant and their interaction partner(s), and *negative* approach/avoidance: behaviors that do not seem conductive to relationship maintenance. For each combination of reaction (approach vs. avoidance) X relationship (positive vs. negative), we made two items: Nngative avoidance, e.g., "Next time, I would avoid having this conversation altogether" ($r = .85$, $p < .001$), positive avoidance, e.g., "Next time, I would change the topic compared to the last conversation" ($r = .64$, $p < .001$), negative approach, e.g., "Next time, when discussing the same topic, I would make my point(s) more forcefully" ($r = .40$, $p < .001$), and positive approach, e.g., "Next time, I would enter the conversation with an open mind" ($r = .50$, $p < .001$). All items were rated on 5-point Likert scales from 1 = *Strongly disagree* to 5 = *Strongly agree*. We only formulated predictions for negative avoidance intentions because this was the first time that we used these items, and we expected this behavioral intention to show the strongest effects. The other intentions were measured exploratively.

*Discriminant validity*. We included the following four variables deemed of importance for conversational experiences (same as Study 1) in order to establish the scale's discriminant validity, and to include as regression covariates: 1) the number of interaction partner(s) in the conversation (1 = *1 other*, 2 = *2 others*, 3 = *3 others*, 4 = *more than 3 others*), 2) to what extent participants knew the interaction partner(s) before (1 = *Not at all*, 2 = *Barely*, 3 = *A bit*, 4 = *Quite well*, 5 = *Very well*), 3) how positive participants felt about their relationship with the interaction partner(s) in general (*1 = Very negative*, *2 = Negative*, *3 = Neutral*, *4 = Positive*, *5 = Very positive*), and 4) whether participants experienced this relationship as equal or not (1 = *Yes, we are equals*, 2 = *No, the other has more authority (e.g., other is my employer)*, 3 = *No, I have more authority (e.g., other is my child)*, 4 = *Don't know*).

## Results

As in Study 1, participants described a variety of conversations, see S12 Table in S1 Appendix for a characterization. A one-sample t-test showed that the mean level of feeling heard was significantly higher than the scale midpoint ($t(999) = 16.51$, $p < .001$, Cohen's $d = .52$, medium effect). The distribution of feeling heard was again, but less strongly than in Study 1, skewed to the left (skewness = -0.55; $M = 3.51$, $SD = 0.98$, S2 Fig in S1 Appendix). This means that most participants wrote about a conversation in which they did feel heard.

**Reliability.** We performed a maximum likelihood confirmatory factor analysis on the eight feeling heard items to test Hypothesis 1. As in Study 1, we used the lavaan package (version 0.6–9; [43]). The model was specified in line with Study 1: the eight items loading on a single factor with a residual correlation between the two negative items. This resulted in a good model fit: $\chi2(19) = 67.46$, $CFI = .99$. The factor loadings ranged between |.67| (for the reverse coded items) and |.89| (for the central feeling heard item), see Fig 4. The omega of the scale was .93, indicating high internal consistency. The inter-item correlations averaged |.66| and ranged between $r = |.45|$ and $r = |.79|$, which means they did not exceed the item exclusion cutoffs used in Study 1. See S13 Table in S1 Appendix for all inter-item correlations as well as the means and standard deviations per item. Taken together, we find strong support for Hypothesis 1: the feeling heard scale is reliable.

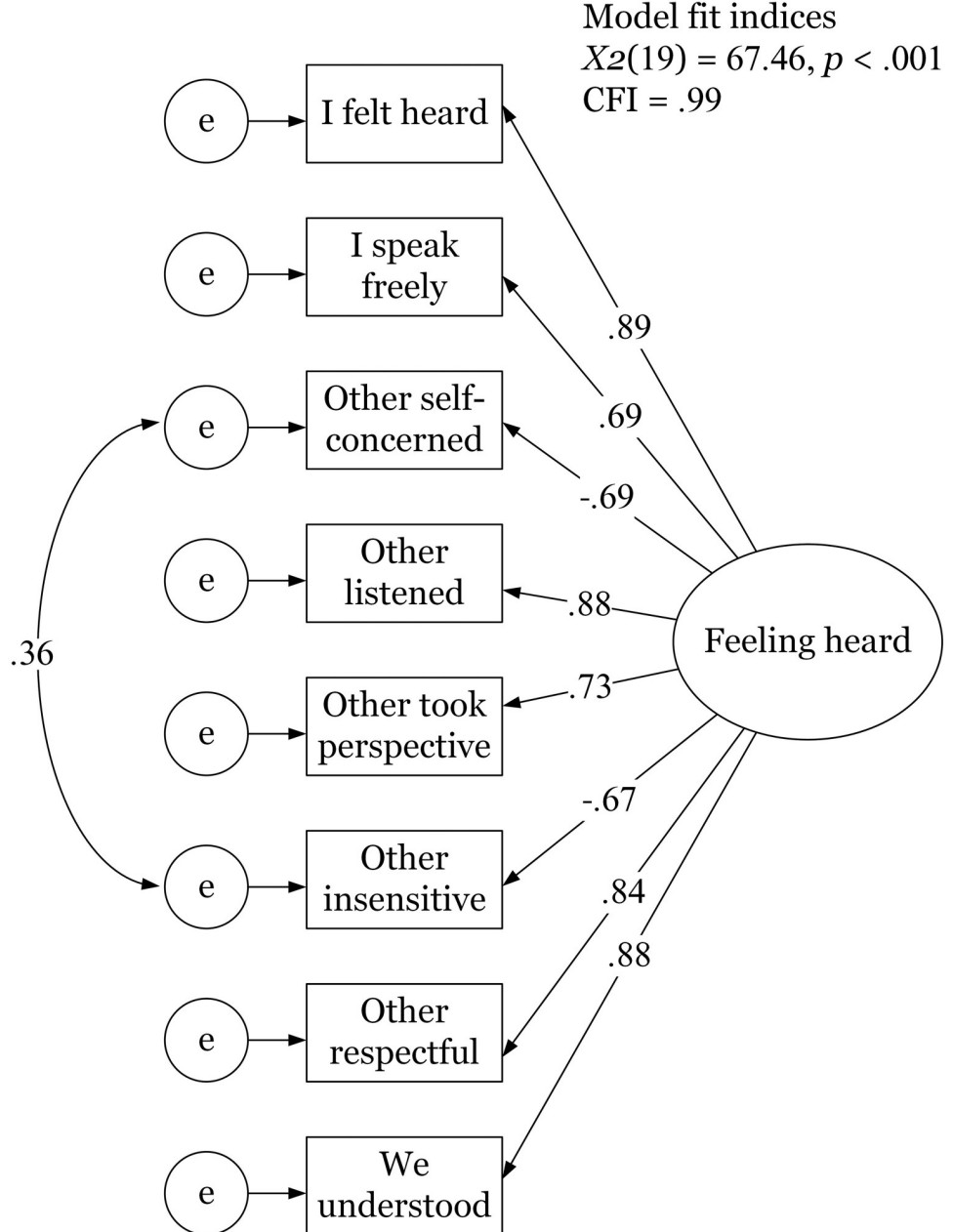

**Fig 4. Factor loadings resulting from the confirmatory factor analysis in Study 2.** *Note.* Rectangles represent the observed variables and the circle represents the latent variable. Standardized factor loadings are displayed.

**Convergent and divergent validity.** To test Hypothesis 2, we calculated the correlations between the feeling heard scale and scales assessing related constructs. As can be seen in Table 5, correlations were significant and in expected directions. Two observations stand out. First, communication apprehension was very weakly related to feeling heard and the other scales. This could be because it is the only scale *not* tapping into evaluations of the specific conversational experience or the specific interaction partner(s), but into participants' attitudes towards conversation in general. Second, the correlation between feeling heard and intimacy

**Table 5. The correlations between the scales as well as their respective means and standard deviations.**

| Variable | Mean (SD) | 1 | 2 | 3 | 4 | 5 | 6 | 7 |
|---|---|---|---|---|---|---|---|---|
| 1. Feeling heard[a] | 3.51 (0.98) | | | | | | | |
| 2. Intimacy[b] | 4.86 (1.32) | .84 | | | | | | |
| 6. Attraction[c] | 6.54 (2.00) | .63 | .72 | | | | | |
| 4. Responsiveness[c] | 5.75 (2.16) | .66 | .76 | .78 | | | | |
| 7. Goal accomplish-ment[a] | 3.37 (1.22) | .75 | .65 | .49 | .51 | | | |
| 3. Dominance[b] | 3.59 (1.10) | -.68 | -.60 | -.54 | -.53 | -.55 | | |
| 5. Distrust[b] | 2.41 (1.31) | -.72 | -.78 | -.80 | -.76 | -.55 | .60 | |
| 8. Communication apprehension[a] | 2.86 (1.04) | -.13 | -.17 | -.12 | -.15 | -.12 | .11 | .11 |

*Note*. All the correlations reported in this table are significant at *** $p < .001$.

[a]Measured on 5-point Likert scales.

[b]Measured on 7-point Likert scales.

[c]Measured on 9-point Likert scales.

exceeded |.80|, which suggests these scales might be measuring the same thing. It is probably no coincidence that conversational intimacy is a subscale of the only scale we could find that, like feeling heard, taps into conversational experiences, albeit intimacy focusses on people's perceptions of their interaction partner's behaviors ("you") while feeling heard also taps into people's internal feelings ("me") and their perceptions of their partner and themselves together ("us").

To check whether feeling heard and intimacy are actually measuring the same underlying construct, we compared the fit of the model where all items of the feeling heard and intimacy scales were set to load on one single factor with the model where feeling heard and intimacy were considered two separate factors. The results show that the model fit is significantly better for the two-factor solution as compared to the one-factor model ($\Delta\chi^2(1) = 804.43$, $p < .001$), but still the two latent factors are strongly related (covariance of .89), which suggests that the constructs might be redundant [57]. Inspecting the single-factor output more closely, reveals why the correlations between constructs and latent factors are so high. The intimacy scale consists of 20 items, with standardized factor loadings ranging from quite low (|.31|) to very high (|.91|). Among the highest loading items are the items which we could also have included in the feeling heard scale, such as "the other person was willing to listen to me" (with a standardized loading of .91) and "the other person was open to my ideas" (.89). And clearly these are things that one might be experiencing in an intimate relationship. But crucially, the items that have greater face validity for intimacy load much less highly, such as "the other person did not want a deeper relationship with me" (-.55), "the other person seemed to care if I liked him/her" (.64), and "the other person was not attracted to me" (-.31). In sum, the feeling heard scale correlates highly with the items on the intimacy scale that also measure experiences very similar to "feeling heard", but the FHS does not correlate highly with the items in the intimacy scale measuring "intimacy". This means that the feeling heard is distinct from intimacy, but the intimacy scale confounds the two and its central dimension (at least in our dataset) is feeling heard. This observation is further supported by the differing correlation patterns of feeling heard and intimacy with other measures, see the section "Feeling heard and intimacy" below.

**Predictive validity.** To test Hypothesis 3, we regressed negative avoidance intentions onto the feeling heard scale; in line with the preregistration, no covariates were included at this point. Feeling heard strongly predicted negative avoidance intentions ($b = -.74$, 95%CI [-.80, -.68], $\beta = -.62$, $R^2 = .38$). The less people felt heard, the more likely they were to say that they were disposed to negatively avoid a next conversation with the same person(s), for instance by

withdrawing from the conversation. To explore the robustness of this effect, we performed a hierarchical linear regression. Block 1 only included the demographic variables as predictors, to which we added the covariates in block 2. In block 3, we added feeling heard, and in block 4 we included all the other scales. Does feeling heard predict intentions to negatively avoid future conversations over and above factors like being acquainted, having a good relationship, or experiencing intimacy?

As can be seen in S14 Table in S1 Appendix, feeling heard was the strongest predictor, after controlling for the influence of all demographic variables and covariates (with $\beta$ = -.54 for feeling heard and $\beta$ between |.01| and |.16| for the other variables). Including feeling heard in the model increased its explained variance from $R^2_{adj}$ = .22 to $R^2_{adj}$ = .42. Importantly, the amount of explained variance only increased by 3% after including the seven scales that measure constructs closely related to feeling heard (from $R^2_{adj}$ = .42 to $R^2_{adj}$ = .45). This means that the other scales do not capture variance in negative avoidance intentions that feeling heard misses. Moreover, feeling heard remained the strongest predictor of negative avoidance intentions with a $\beta$ of -.24, whereas the standardized beta coefficients of the other 15 predictors ranged between |.01| and |.13|.

In additional explorative analyses, we assessed the effect of feeling heard on the other behavioral intention measures, controlling for all demographics and covariates. Although the effects were somewhat smaller, feeling heard significantly predicted positive avoidance ($b$ = -.57, 95%CI [-.65, -.50], $\beta$ = -.50, $R^2$ = .29), negative approach ($b$ = -.22, 95%CI [-.29, -.14], $\beta$ = -.21, $R^2$ = .11) and positive approach intentions ($b$ = .29, 95%CI [.23, .35], $\beta$ = .33, $R^2$ = .28), see S15 Table in S1 Appendix.

The central item of the scale (i.e., "In this conversation, I felt heard by the other(s)") also strongly predicted behavioral intentions. Controlling for the demographics and covariates, the unstandardized betas of this item ranged between |.12| (negative approach) and |.41| (negative avoidance) with explained variances from 9% to 37%. Similar to Study 1, the item-total correlation of this central item was .71, meaning that it can account for 50% of the variance measured with the full scale. This suggests that this single item might be a good short alternative for the full scale.

Thus, we find support for Hypothesis 3 and show that feeling heard is a distinct and powerful predictor of post-conversation avoidance intentions. Besides being a strong predictor, feeling heard explained variance over and above 15 related constructs. This again shows that feeling heard means something else than experiencing intimacy, dominance, distrust, liking, etc.

**Feeling heard and intimacy.** An important plausible cause for the high correlation between feeling heard and conversational intimacy is the fact that most of the sample described what can be considered an intimate relationship: more than 80% felt positive to very positive about the relationship with their interaction partner(s).

Whereas intimacy should be relevant and predictive of intentions among close others, feeling heard should be predictive in all subsamples. We tested this in two subsamples: 1) participants that felt not or only moderately positive about their relationship ($N$ = 192), and 2) participants that felt positive or very positive about their relationship ($N$ = 808).

First, in the "non-positive" subsample, when both constructs were simultaneously included in a multiple regression, feeling heard was still strongly predictive of negative avoidance intentions ($\beta$ = -.49) while intimacy was not ($\beta$ = .02). In contrast, in the "positive" subsample, both constructs maintained predictive value: $\beta$ = -.35 for feeling heard and $\beta$ = -.26 for intimacy. Thus, whereas conversational intimacy only predicted negative avoidance intentions when people felt positive about their relationship, feeling heard was also predictive in the context of less positive relationships. We can conclude that, especially in this context, feeling heard and intimacy are not the same.

**Discriminant validity.** Lastly, to establish discriminant validity, we explored the correlations between feeling heard and other variables relevant for conversational experiences. Feeling heard was not related to the number of interaction partner(s) in the conversation ($r = -.04$, $p = .178$) and to whether people knew these interaction partner(s) before the conversation ($r = .08$, $p = .014$). In contrast, feeling heard was strongly positively related to the perceived quality of this relationship ($r = .52$, $p < .001$). Moreover, participants felt more heard when they described a conversation with other(s) of equal status than when they considered either the other(s) or themselves of greater authority, or when participants indicated they did not know whether their relationship was equal or not ($F(3,996) = 15.211$, $p < .001$; $M_{equal} = 3.63$, 95%CI [3.56, 3.71] vs. $M_{other-authority} = 3.24$, 95%CI [3.11, 3.37], Cohen's $d = .41$, small effect, $M_{self-authority} = 3.25$, 95%CI [3.09, 3.61], and $M_{don't-know} = 2.82$, 95%CI [2.48, 3.16], Cohen's $d = .31$, small effect). The demographic variables (age, gender, education, and employment status) did not significantly relate to feeling heard. We thus replicate the results of Study 1 regarding the correlations between feeling heard and these conversational variables as well as the demographics. The finding that feeling heard is not related to being acquainted suggests that the scale has discriminant validity.

## Discussion

Study 2 shows that the feeling heard scale is reliable and has high internal consistency, not only in the context of online interactions but in conversation in general. These results are important since Study 2 employed a distinct sample in a different cultural setting as compared to Study 1 and we see that the scale performed in substantially identical ways in both (Dutch and US) samples. The scale further showed convergent and divergent validity: feeling heard is related to, but different from, conversation behaviors that communicate intimacy and dominance, perceived responsiveness of interaction partner(s), attraction and distrust towards interaction partner(s), and the experience of having accomplished conversational goals. Discriminant validity was evidenced by the lack of a relation between the FHS and pre-conversation acquaintance. Lastly, we found that the scale is a distinct and strong predictor of conversational intentions related to conflict. Not feeling heard in a conversation can motivate people to avoid another conversation with the same partner(s), and, conversely, the experience of feeling heard makes it more likely that people will enter a next conversation with an open mind. The latter observation is in line with recent research showing similar effects for high quality (attentive, empathic and nonjudgmental) listening [22, 58].

## General discussion

Based on a comprehensive literature review, expert interviews, and scale validation studies, we define feeling heard as a concept covering the three entities in interaction: feeling heard is about me, about you, and about us. We distinguished five components of feeling heard—voice, attention, empathy, respect, and common ground—based on the literature review. Empirically these loaded on one single factor. This suggests that feeling heard is a unitary concept, an unidimensional feeling (see [49] for a similar result concerning perceived listening). We then developed and validated a reliable and concise eight-item scale to measure it: the Feeling Heard Scale (the FHS).

### Feeling heard matters across contexts

We found that feeling heard is relevant in many different contexts and relationships. When respondents were asked in an open-ended manner to describe situations in which they felt heard or not, they reported these feelings in conversations ranging from insurance

negotiations, fobbing off an advertiser, or business meetings, to exchanging intimate details with friends or catching up with distant family. In both Study 1 and 2, feeling heard was unrelated to acquaintance, meaning that feeling heard (or not) is experienced in encounters with close friends as much as with strangers.

Across different contexts and relationships, we find that when people reflect on a significant social interaction they recently had, they are more likely to report that they did feel heard than did not feel heard. Although we cannot say this with certainty, this suggests that across many individuals, there might be a base rate for people to feel heard in most of their conversations. If this also holds within individuals, this may make not feeling heard a negative expectancy violation which could lead to negative arousal (see also [52]). By extension, feeling heard might play an especially central role in contexts where people more often do not feel heard. Situations in which people tend to feel less heard are also quite diagnostic and can aid our understanding of what feeling heard (and not heard) entails psychologically and socially. For example, our research suggests that these can be situations of unequal status.

Both studies showed that participants felt less heard when they were in a subordinate position and when they felt their interaction partner was dominating the conversation. A dominant other tends to hold the floor and is likely to be somewhat less concerned with others. This may mean that there is less room for the participant to express (voice) and makes it less likely that the participant feels attended to and respected. Interestingly however, people who occupied the higher status position in the conversation they reported about, did not feel *more* heard than those reporting about being in the subordinate position. This might be because in such an interaction, the subordinate other cannot reciprocate since they themselves do not feel heard—in this sense, it may indeed be lonely at the top. All this is in comparison to equal status interactions, where people tend to feel most heard. So, it appears that in unequal status interactions both sides lose out, and that the optimal situation is when both parties are on equal footing. This finding reinforces our impression that feeling heard could be seen as a product of a *successful collaboration* in conversation. Thus, these and other situations in which feeling heard tends to be most threatened (e.g., online conversations) underline the defining components of feeling heard and can provide additional insights into their workings.

## The FHS is a strong predictor

We found that feeling heard is a distinct and powerful predictor of conversation intentions related to conflict. The FHS is an especially strong predictor of intentions to engage in another conversation with the same partner(s), univariate as well as when many other relevant predictors are included. In both cases the FHS explains around 30 percent of variance in conversation avoidance intentions. The FHS explains a similar amount of variance in intentions to enter a next conversation with an open mind.

This echoes research showing that perceived responsiveness and listening can lessen defensive reactions [16, 58, 59]. Note that this is not about being in agreement, but about the way people discuss about their *dis*agreements. In fact, the item implying disagreement "we had a different perspective" did not fit the FHS well, and disagreement was barely mentioned in the open answers. This suggests that people can disagree but still feel heard. Feeling heard could therefore help bridge social divisions within society, whilst maintaining a plurality of voices. There is already evidence that feeling understood by an outgroup can have these positive effects, such as increasing trust and forgiveness, e.g., [59, 60].

Notably, feeling heard was the strongest predictor of negative avoidance intentions across a broad range of contexts and relationships, explaining variance over and above 15 related variables, such as interpersonal acquaintance. The FHS might be such a good predictor, because,

in the way we operationalized it, feeling heard is not just an assessment of the relational qualities of self, other, *or* the relationship (as most variables in the interpersonal relations literature tend to be), but rather a variable that reflects the social interaction and is therefore about me, you, and us, all at the same time. For instance, the FHS was predictive in the subsamples of participants who reported about interactions with both people they were positive and less positive about, while the closely related conversational intimacy scale was only predictive in interactions with others who participants were initially positive about. Conversational intimacy therefore seems to be better equipped to operationalize aspects of the quality of "strong tie" relationships, whereas the FHS is a good predictor in conversations with partners who have either strong or weak ties. Of course, intentions are held in the present and do not always translate into actual behavior (i.e., the intention-behavior gap). It is therefore imperative that future studies map the behavioral consequences of feeling heard experiences over time to further establish the scale's predictive validity.

## Practical recommendations

Feeling heard can be considered a process variable. It is shaped by and shapes interaction dynamics. Whether one feels heard depends on the behaviors of one's interaction partner(s) and one's attributions and perceptions of these behaviors. The degree to which one feels heard then shapes one's behaviors towards these partners who, in turn, interpret and respond to one's behaviors. This focus on conversational experiences is also what makes the FHS unique; most related scales measure general perceptions and feelings towards either the self, partner(s), or relationship.

In many contexts, to fully understand (and predict) the consequences of certain interaction behaviors, the valid assessment of feeling heard can thus be essential. Beyond the contexts in which feeling heard is known to play a role (intimate relationships, healthcare, work), the FHS can be relevant in any context that involves direct interpersonal interaction. In therapeutic contexts, the FHS could help understand why certain treatments might be more (or less) effective; knowing whether a client feels heard by their therapist might be essential to understand the effectiveness of certain listening techniques. Call centers or helpdesks could monitor customer satisfaction by assessing the extent to which customers feel heard. In ongoing relationships, the FHS may be found suitable as an instance-by-instance measure to track patterns of relationship development and maintenance. At a different level, feeling heard might also have an important role to play in the relationship between governmental institutions and citizens: whether citizens that are affected by a certain governmental decision feel heard in their contact with governmental representatives might predict their acceptance of that decision (similar to voice, [30]).

It holds for all these settings that now we can measure feeling heard, we can test the assumptions surrounding this construct: how important is feeling heard in a specific context? What makes people feel heard in conversation? What kinds of processes are mediated by feeling heard? Instructions on how to use the Feeling Heard Scale can be found in the user manual in section 4 of the S1 Appendix. Please note however, that further research is needed to validate and standardize the FHS in clinical and other applied settings, as well as other populations, such as children and respondents with a non-western cultural background.

The FHS is a relatively compact and effective measure to use in most settings, but in research environments where space is at a premium, we recommend using the central item: "In this conversation, I felt heard by the other(s)". Of course, there is always a loss of precision when using a single item, but on the basis of our research we can quantify its magnitude. Both studies show that the amount of variance in the full scale that is accounted for by the single-

item alternative is about 50%. If one chooses to use the single item, one should be mindful of this loss of precision, and transparent about it in communicating its results. Larger sample sizes could compensate for this, but in a clinical setting or other context where $N = 1$, we strongly recommend the use of the full scale for a reliable and precise assessment of feeling heard.

Finally, our theoretical framework provides guidelines on how to increase interaction partners' experiences of feeling heard. Depending on the feeling heard component(s) one decides to tackle, the commitment of one to three entities is needed: the speaker, the listener, and/or both of them together. Can the person that does not feel heard make sure they can have their say? This would depend importantly on the exact source of the silencing: maybe the speaker should speak louder or explicitly ask for attention. Does the person that wants to make another feel heard show enough attention, empathy, and respect? Here active listening techniques could be of help, e.g., paraphrasing, open-ended questions [23]. Or do both parties need to work on establishing a sense of common understanding? This could be done by common grounding techniques, e.g., checking for understanding and asking for clarification [35].

### Limitations and future directions

The current studies showed that feeling heard is related to the quality (not to be confused with quantity: acquaintance) and equality of the relationship. But the direction of these relationships is not clear. It might be a bi-directional process in which "I feel positive/ equal towards the other(s) because I feel heard, and I feel heard because I feel positive/ equal towards the other(s)". Feeling heard might affect and/or be affected by many more dispositional (e.g., rejection sensitivity, self-esteem), situational (e.g., expectations), and cultural (e.g., norms) factors. For instance, people that are low in self-esteem or especially sensitive to signals of rejection might tend to feel not heard more easily (such effects have been observed for ostracism, see [61]). As an example of cultural influences, a Turkish-Dutch scientist we interviewed mentioned that the right to be (and feel) heard needs to be earned in more collectivist cultures (see section 1.3 in S1 Appendix). This is related to the relatively limited demographical diversity in our sample (often younger adults, highly educated, likely relatively affluent and mostly white). These are all interesting directions for future research.

The current paper looked at feeling heard in conversations. However, as alluded to previously, the experience of feeling heard may also exist at a more abstract level, without direct interpersonal interaction. One prominent example that comes to mind is reflected in the quote by Dr. Martin Luther King at the start of this paper: citizens feeling not heard by government. The consequences of this experience are well-documented. For example, political discontent tends to be highest among less educated people because they feel politicians do not truly understand their lived experiences and do not respect them [62]. Moreover, qualitative research suggests that not feeling heard plays an important role in the reasons for activists to step up their actions, become more vocal, and ultimately to embrace more radical forms of collective action [6]. Future empirical investigation is required to establish the nature of feeling heard at the abstract level and its relationship to feeling heard in conversations. It might be that feeling heard in conversations with representatives of an institution or another group can spill over to feeling heard by that institute or group as a whole.

### Conclusion

In many, if not all, interpersonal contexts, from work meetings and customer support to parent-child interactions and therapeutic contacts, the experience of feeling heard reflects an essential component of social interaction. By developing and validating a scale to measure

feeling heard, this paper opens up the possibility to study this key process variable. The central conceptual contribution of this research is that it shows that feeling heard is not about any individual interaction partner, nor is it only about the collective, nor is it just about the appreciation of the interaction itself. Feeling heard is about all these things together, at the same time. This concept reflects the interpersonal and social *dynamics* of conversation, as they constitute the relational *outcomes* for self, other, and for "us". We believe that in individualized societies, where the sense of self occupies an increasingly central role, this concept may become more important not just for individual self-esteem and well-being and for high quality relations, but more generally for maintaining the social fabric of societies.

## Supporting information

**S1 Appendix. The supporting information file containing 1) all supporting figures, tables, questionnaires and additional qualitative data of the two studies reported in the paper, 2) the report of a third (lab) study, and 3) the user manual of the feeling heard scale.** (DOCX)

## Acknowledgments

The authors thank the four anonymized interviewees for sharing their knowledge about and experience with feeling heard from a practical perspective. The authors additionally thank Angela Voskuilen for performing the double-coding of the qualitative data of Study 1. Study 3 in the S1 Appendix was made possible by Kira Choinski who co-designed the experiment and collected the data. Lastly, the authors are grateful to dr. Paul Vermeulen and dr. Dominik Neumann for their advice concerning conceptualization and operationalization.

## Author Contributions

**Conceptualization:** Carla Anne Roos, Tom Postmes, Namkje Koudenburg.

**Data curation:** Carla Anne Roos.

**Formal analysis:** Carla Anne Roos.

**Funding acquisition:** Carla Anne Roos, Tom Postmes, Namkje Koudenburg.

**Investigation:** Carla Anne Roos, Tom Postmes, Namkje Koudenburg.

**Methodology:** Carla Anne Roos, Tom Postmes, Namkje Koudenburg.

**Project administration:** Carla Anne Roos.

**Resources:** Carla Anne Roos, Tom Postmes, Namkje Koudenburg.

**Software:** Carla Anne Roos.

**Supervision:** Carla Anne Roos, Tom Postmes, Namkje Koudenburg.

**Validation:** Carla Anne Roos, Tom Postmes, Namkje Koudenburg.

**Visualization:** Carla Anne Roos.

**Writing – original draft:** Carla Anne Roos, Tom Postmes, Namkje Koudenburg.

**Writing – review & editing:** Carla Anne Roos, Tom Postmes, Namkje Koudenburg.

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
