## [Decision Letter · Decision Letter 0]

15 Aug 2023

PONE-D-23-13419Feeling heard: Operationalizing a key concept for social relationsPLOS ONE

Dear Dr. Roos,

Thank you for submitting your manuscript to PLOS ONE. After careful consideration, we feel that it has merit but does not fully meet PLOS ONE’s publication criteria as it currently stands. Therefore, we invite you to submit a revised version of the manuscript that addresses the points raised during the review process.

We look forward to receiving your revised manuscript.

Kind regards,

Ramona Bongelli, Ph.D.

Academic Editor

PLOS ONE

Journal Requirements:

” Part of this research (i.e., Study 2) was funded by the Behavioral and Social Sciences Internet Research Fund of the University of Groningen.”

“Part of this research (i.e., Study 2) was funded by the Behavioral and Social Sciences Internet Research Fund of the University of Groningen (project code 170240169). The grant holder was Namkje Koudenburg; Tom Postmes and Carla Roos were co-applicants. The funder's website is https://www.rug.nl/gmw/?lang=en

The funder had no role in study design, data collection and analysis, decision to publish, or preparation of the manuscript.

No further grant was received from any funding agency in the public, commercial, or not-for-profit sectors.”

Reviewers' comments:

Reviewer's Responses to Questions

**Comments to the Author**

1. Is the manuscript technically sound, and do the data support the conclusions?

Reviewer #1: Yes

Reviewer #2: Yes

2. Has the statistical analysis been performed appropriately and rigorously? 

Reviewer #1: Yes

Reviewer #2: Yes

3. Have the authors made all data underlying the findings in their manuscript fully available?

Reviewer #1: Yes

Reviewer #2: Yes

4. Is the manuscript presented in an intelligible fashion and written in standard English?

Reviewer #1: Yes

Reviewer #2: Yes

5. Review Comments to the Author

Reviewer #1: This manuscript provides a substantial review of the literature to make a compelling case for exploring and measuring “feeling heard”. It then reports two empirical studies describing the development and validation of the Feeling Heard Scale (FHS). I believe the manuscript makes a strong contribution to the extant literature, and I expect the FHS will be well-received and widely used in distinct research domains (e.g., intimate relationships, communication). However, there are some issues for consideration. Here I offer few suggestions for the authors to make their contribution stronger.

1. I understand the goal of testing the conceptual model developed in Study 1 and use the opportunity to report qualitative insights from participants. But I wished the authors would have conducted two distinct studies: the first a more qualitative study investigating how the feeling heard conceptualization made by the authors matched respondents’ own understandings and experiences; and the second study then updating the conceptualization of the theoretical model based on the first study to then guide the development of the measure. This combination of deductive-inductive-(and back to)deductive approach would have made the empirical work stronger, and potentially the final measure. I therefore recommend the authors to mention this as a potential limitation of their work.

2. I also have some recommendations regarding the theoretical model. I really liked the five identified facets (i.e., voice, attention, empathy, respect, and common ground), their clear grounding in the literature (which could have been improved by initial matching text with respondents; see above), and clear rational for the specific items in Table 1. However, aspects of the model conceptualization and presentation can be improved. First, I find the distinction between interpersonal and collective confusing. Common group (‘we’) is clearly within an interpersonal domain, and the word ‘collective’ has a much broader and group-based meaning that goes beyond ‘we’, so reading this word makes the model confusing. Third, the model description implies a hierarchical conceptualization – interpersonal (‘me’ and ‘you’) nested within ‘collective’ (‘we’) – which is not empirically explicit/tested by the model. Finally, the model conceptualization in Figure 1 does not follow recommended practices.

So, I would recommend the following: (a) I would avoid the use of ‘collective’ in the model. Perhaps ‘collective’ could be replaced with ‘dyad’, or perhaps better use ‘unipersonal’ and ‘interpersonal’ to making the distinctions between ‘me/you’ and ‘we’, respectively. (b) I would check the model description in text to avoid potential reference to a hierarchical model. This has two potential understandings: one is a multilevel model (e.g., students within classrooms within schools; or ‘me’ and ‘you’ within ‘we’) and another is a hierarchical factorial model (i.e., the five facets of voice, attention, etc nested within the higher-order construct of feeling heard) – but neither of these models were tested empirically, so I would avoid any hierarchical description in the text. And (c) I recommend the authors to use an oval for “feeling heard” in Figure 1 as this is the convention for depicting latent constructs.

3. Importantly, the correlations reported in Table 5 clearly indicates non-trivial conceptual overlaps between feeling heard and neighboring constructs. The authors provide some examination for the overlap with intimacy (correlation of .84!), but I do believe the authors must explore this further. In particular, I recommend the authors to read carefully the recent paper by Gordon Hodson titled "Construct jangle or construct mangle? Thinking straight about (nonredundant) psychological constructs” (https://doi.org/10.1002/jts5.120). For example, I do think the authors could provide further evidence these constructs are empirically distinct via examining the correlations of the latent factors using structural equation modelling. I understand the authors want to promote their new construct and scale, but the authors must provide convincing and (to same extent) exhaustive that this construct is justifiable non-redundant. In my view, one aspect of the Open Science movement that the authors follow (e.g., pre-registering Study 2, making material/data available) is to avoid ‘noise’ within psychological science, and promoting redundant measures is an example of such noise. This would only require the authors to reanalyze their existing data (and not to collect more data) and the work is not considerable, but the findings might go against the initial predictions (and desires) of the authors which should be accepted, and which is more aligned with the scientific enterprise (e.g., focus on the evidence and not on individual preferences).

4. There are also smaller changes, including the following:

- perhaps report one-sample t-tests when reporting mean levels of feeling heard to indicate whether it differs from the scale middle point

- in case the authors retain the additional intimacy analysis reported in p. 35, perhaps they could examine this more formally by running a moderation analysis with sample type as the moderator; this has the added benefit of avoiding splitting the sample variable.

- It is unclear to me how there were “214 participants for Part B” as reported in p. 26. Please double check this. I also found the description of the conditions and two parts in p. 13 confusing, and you might want to improve it.

- It is unclear to me the interpretation that “all items tapped into feeling heard” based on the t-test reported in p. 22

- The assertion that item 1 “loads somewhat on various factors” is misleading as Figure 1 clearly indicates it loads reliably only on two factors (respect and common group).

- I also noticed some typos/grammar issues (e.g., “participants / others that” instead of “participants / others who”; “study 1”, inconsistency in using “confirmatory factor analysis” and COVID with initial capital or not; sentence with many ‘understanding’ in line 222)

Reviewer #2: This study sought to develop a scale to measure feeling heard (FHS). Feeling heard is described as a complex construct of five components at two levels: at the interpersonal level, having voice; receiving attention, empathy, and respect; and at the communal level, experiencing common ground. These are described as involving me, you, and we.

Content

1. The abstract does not identify the two levels: interpersonal and collective. It seems this would be important to the reader, so state this in the abstract. Perhaps more problematic, the abstract currently seems to imply two sets of five elements (“five components at two levels”) and the two levels suggests higher-order factor structures while two conceptual levels are in fact intended. But there are actually three personal contexts: me, you, and us. So this may be better described as three interpersonal contexts: “individuals feel heard when 1) my voice receives 2) your attention, 3) empathy, and 4) respect in the context of 5) our common ground.

2. A key point in the discussion—lines 571 and 832—is that “feeling heard appears to be a unitary concept.” That needs to be said in the abstract as well. At present the abstract leaves the impression that feeling heard and the resulting FHS is five-dimensional.

3. Figure 1 line 234. It seems oddly inconsistent and a bit disconcerting to insert tables into the text and place figures following the references—see note #1 under Style below.

4. Implicit in Study 2 is the belief that the scale will perform in substantially identical ways in the Dutch and US samples. That it does should be explicitly stated in the discussion; that is an important finding.

5. Line 246 should conclude “can be found in the . .”

6. Informed consent could not be examined due to masking materials. It seems a better plan is needed to support review of this feature of the study. Briefly describe the consent procedures for each set of data.

7. The 2 x 2 design is not clear. What steps were taken to ensure that the same conversation was not reported in Part A and Part B?

8. Rather than state you “tried to include one negatively phrased item,” report the number of negatively-worded items in the initial and final FHS scales. The insertion of a correlation between negatively-worded items suggests there is a method effect for negatively-worded items. That might be good to include in the discussion. There is a growing body of evidence that negatively-worded items do not work similarly across some cultural settings (e.g., see Bufford & Paloutzian, 2023).

9. The section about descriptions and reasons—lines 342-344—seems redundant. Or perhaps clarification is needed.

10. Line 347 might read better as the average of both coders or the mean score of both coders.

11. Perhaps it would be worth noting in the discussion that underlying being heard are a set of shared values: respect for the person, sharing of power, a sense of justice carried out, common ground/worldview, shared being (me, you, us). And perceptions of this mutuality is essential to feeling heard.

12. It would help to carry through with the enumeration begun in line 359. What comes second and third?

13. Perhaps add here that over half of the remembered Last conversations were with colleagues and family.

14. Table 3 needs a bit more detail. For percentages, it is not clear what the underlying referent is.

15. Line 457 is not a complete sentence. Perhaps combine with the previous sentence.

16. At lines 468 to 470 absolute values of t would be clearer. As stated, a t of zero is included as significant.

17. I suggest stating that confirmatory factor analysis rather than exploratory factor analysis was carried out because of the theoretical model for feeling heard that shaped the study.

18. Given the length and complexity of the supplementary materials, at times the reader needs more help in navigating them. A table of contents at the front of the supplementary material might work. Or refer to the page(s) on which the supplemental material is presented. A header and pagination are also essential for this material.

19. Effect size data are needed for the analyses of variance at lines 545-548.

20. Limitations of Study 1 are not mentioned in the discussion. An important one is that the test for generality of the measure across samples (“invariance”) is quite weak as the same sample was employed: generality is only assessed across conversation settings within the same sample.

21. The reference to procedural and distributive justice would benefit from a brief explanation—an appositive—of these concepts (justice processes and outcomes?) and their relevance to feeling heard.

22. At line 589 place convergent and divergent validity together rather than separating them with predictive validity. Placing predictive validity last would better parallel the following hypotheses as well.

23. The divergent validity hypothesis at line 608 is problematic. First, it purports to test the null hypothesis of no relationship. Second, divergent validity is more accurately a negative relationship rather than the absence of a relationship. Discriminant validity? Hypothesis 2B seems to best fit divergent validity as I understand it. A pertinent methodological citation(s) may be helpful as these concepts seem to be defined and used in inconsistent ways.

24. The intentions for a next conversation measure is gathered as concurrent data: intentions are held in the present. Considering it a measure of predictive validity stretches the concept. 675-678

25. A period seems to be in the wrong place at line 686. Actually several; check punctuation in this entire paragraph.

26. The data at 679-689 lack needed context. What are these relationships?

27. It seems unclear how the variables following 693 fit the concept of divergent validity. Perhaps a technical citation is needed here as it appears the literature seems to conflate divergent validity and discriminant validity. See #23 above.

28. The CFA for Study 2 is much more valuable than that for Study 1 as it employs a distinct sample in a distinct cultural setting. This should be mentioned in the discussion of Study 2 or General Discussion.

29. Table 5 would be more effective if it were constructed to parallel the hypotheses 2A and 2B. The present format makes the reader work to follow the expected direction of relationships and minus signs are scattered rather than clustered together.

30. Line 735 claims that “feeling heard differs from intimacy”, but it appears this claim was not statistically tested—are these correlations significantly different? Perhaps simply report they show somewhat different patterns of correlation with other measures.

31. While the logic of regression is predictive, data were gathered at the same time and is somewhat misreported at line 745. It would be more accurate to say “The less people felt heard, the more likely they they were to say that they were disposed to negatively avoid a next conversation. . . “

32. Taken as a whole the regressions establish the FHS as a predictor of engaging/avoiding conversation and item 1 is a good short version. And yes, feeling heard is distinct from the related constructs. 778-779

33. The claim that “not feeling heard appears to be somewhat less common than feeling heard” is not tested in the study (line 845). It seems all you could claim here is that participants were more likely to report an episode of feeling heard. Might that be due to the relative rarity of feeling heard? Could is be influenced by the initial instructions?

34. At line 891-896 the supporting data are not clear. Does the conversational intimacy comment pertain to a finding I overlooked—or to another study which should be cited here?

35. Characteristics of the ample seem additional limitations—young adults, more educated, likely more affluent, likely mostly white—I did not see any ethnic/racial description of the samples.

36. The supplementary material is a bit chaotic. Adding pagination, a table of contents, and page breaks that fit the content would help to make this material (all 51 pages) accessible.

B. Format/Style

1. In APA style, tables and figures can be embedded in the text or aggregated after the reference list. Here neither of these options is employed; rather they are split in both places.

2. See line 288—in APA style numbers are spelled out to begin a sentence.

3. Embedded tables should be preceded and followed by an extra double line-space (e.g., 354-358).

4. Single-spaced tables are acceptable in APA style, but an additional line space between sections would make the material more readable (e.g., Table 2, 356ff).

5. Correlational tables are generally constructed with the data on the lower left part of the table rather than the upper right half.

6. Figures require a descriptive title and sufficient information in the notes to explain them independent of the text. The header should remain constant except for the page #. Figure titles are placed below the figures. For example, lines 531-536 should be included on the page with the figure. In this manuscript they appear to be inserted into the text without the needed extra spacing to distinguish them from the text.

7. Table S-2 would be more accessible if t and its df were placed in adjacent columns or t and df were listed sequentially like M, (SD). Several of the supplementary tables would also benefit by putting additional material in the same or an adjoining column rather than between rows.

6. PLOS authors have the option to publish the peer review history of their article (what does this mean?). If published, this will include your full peer review and any attached files.

Reviewer #1: No

Reviewer #2: No

---

## [Author Response · Author response to Decision Letter 0]

14 Sep 2023

Reviewer #1: This manuscript provides a substantial review of the literature to make a compelling case for exploring and measuring “feeling heard”. It then reports two empirical studies describing the development and validation of the Feeling Heard Scale (FHS). I believe the manuscript makes a strong contribution to the extant literature, and I expect the FHS will be well-received and widely used in distinct research domains (e.g., intimate relationships, communication). However, there are some issues for consideration. Here I offer few suggestions for the authors to make their contribution stronger.

1. I understand the goal of testing the conceptual model developed in Study 1 and use the opportunity to report qualitative insights from participants. But I wished the authors would have conducted two distinct studies: the first a more qualitative study investigating how the feeling heard conceptualization made by the authors matched respondents’ own understandings and experiences; and the second study then updating the conceptualization of the theoretical model based on the first study to then guide the development of the measure. This combination of deductive-inductive-(and back to)deductive approach would have made the empirical work stronger, and potentially the final measure. I therefore recommend the authors to mention this as a potential limitation of their work.

We agree with the reviewer that two separate studies would have been better, but time and material constraints prevented us from doing so. There are two reasons why we think this approach has not been problematic for the current study: First, we see that the qualitative results show strong support for out theoretical model. Participants’ understanding of feeling heard deviated from our model only on the theme of effort or goal accomplishment. Second, Study 2 was (amongst others) designed to test whether effort/goal accomplishment is a central component of feeling heard, which appeared not to be the case; feeling heard does not necessarily involve effort or goal accomplishment. We now describe this potential limitation in the Discussion section of Study 1, lines 590-593:

“We additionally performed a qualitative analysis of participants’ descriptions of feeling heard. Although this combination of model testing (quantitative data) and generation (qualitative data) in a single study can be seen as a limitation, the derived themes overlapped to a great extent with our definition.”

2. I also have some recommendations regarding the theoretical model. I really liked the five identified facets (i.e., voice, attention, empathy, respect, and common ground), their clear grounding in the literature (which could have been improved by initial matching text with respondents; see above), and clear rational for the specific items in Table 1. However, aspects of the model conceptualization and presentation can be improved. First, I find the distinction between interpersonal and collective confusing. Common group (‘we’) is clearly within an interpersonal domain, and the word ‘collective’ has a much broader and group-based meaning that goes beyond ‘we’, so reading this word makes the model confusing. Third, the model description implies a hierarchical conceptualization – interpersonal (‘me’ and ‘you’) nested within ‘collective’ (‘we’) – which is not empirically explicit/tested by the model. Finally, the model conceptualization in Figure 1 does not follow recommended practices.

So, I would recommend the following: (A) I would avoid the use of ‘collective’ in the model. Perhaps ‘collective’ could be replaced with ‘dyad’, or perhaps better use ‘unipersonal’ and ‘interpersonal’ to making the distinctions between ‘me/you’ and ‘we’, respectively. 

A. We deliberately chose the term “collective” to refer to participants’ sense of themselves and the other as one indistinguishable unit or entity. Also see the literature on collective intentionality (Searle, 1990). We thus do not understand the “we/us” in our model as interpersonal and/or dyadic. By using “collective” and “we/us”, we try to emphasize how feeling heard goes beyond the interpersonal experiences of interaction partners as individuals. We therefore decided to keep this terminology, but clarified the reasoning behind this in the manuscript, by adding a short explanation on lines 183-186:

“We refer to this level of “us” or “we” as the “collective level” to distinguish it from interpersonal experiences. At the collective level interaction partners do not think about themselves and others as individuals but as one irreducible entity (also see the literature on collective intentionality, [38]).”

Searle, J. R. (1990). Collective Intentions and Actions. In P. Cohen, J. Morgan, & M.E. Pollack (Eds.), Intentions in Communication (pp. 401-415). MIT Press.

(B) I would check the model description in text to avoid potential reference to a hierarchical model. This has two potential understandings: one is a multilevel model (e.g., students within classrooms within schools; or ‘me’ and ‘you’ within ‘we’) and another is a hierarchical factorial model (i.e., the five facets of voice, attention, etc nested within the higher-order construct of feeling heard) – but neither of these models were tested empirically, so I would avoid any hierarchical description in the text. 

B. We thank the reviewer for making this important point. The comments made us realize that the interpersonal versus collective “levels” phrasing can be confusing because we indeed do not distinguish between them in our analysis. We therefore followed the suggestion, adapted our model and de-emphasize the collective-interpersonal distinction. We kept the presentation of five components that concern self, other and us. We did so because, whereas the five components that we based on theoretical grounds could also not be distinguished on empirical grounds (feeling heard being a unitary concept), we did test this model in Study 1 (and indeed this is how it was initially conceived and pre-registered). 

Furthermore, as indicated in response to comment 1 by reviewer 2, we replaced the confusing description of “levels of analysis” with “conceptual levels” which is more fitting and clear, we think.

These changes concern multiple sections of the manuscript. But the most significant and important change is in the ”A definition of feeling heard and an operational model” paragraph in the Introduction where we outline our model, see lines 228-232:

“Putting these different insights together, feeling heard has five components concerning three entities (me, you, and we). In order to feel heard, people take into account their interpersonal level experiences of my voice (1), and your attention (2), empathy (3) and respect (4). But beyond that, they may also factor in the collective level experience that we have (5) common ground.”

And (C) I recommend the authors to use an oval for “feeling heard” in Figure 1 as this is the convention for depicting latent constructs.

C. We agree with the reviewer and changed Figure 1 accordingly. 

3. Importantly, the correlations reported in Table 5 clearly indicates non-trivial conceptual overlaps between feeling heard and neighboring constructs. The authors provide some examination for the overlap with intimacy (correlation of .84!), but I do believe the authors must explore this further. In particular, I recommend the authors to read carefully the recent paper by Gordon Hodson titled "Construct jangle or construct mangle? Thinking straight about (nonredundant) psychological constructs” (https://doi.org/10.1002/jts5.120). For example, I do think the authors could provide further evidence these constructs are empirically distinct via examining the correlations of the latent factors using structural equation modelling. I understand the authors want to promote their new construct and scale, but the authors must provide convincing and (to same extent) exhaustive that this construct is justifiable non-redundant. In my view, one aspect of the Open Science movement that the authors follow (e.g., pre-registering Study 2, making material/data available) is to avoid ‘noise’ within psychological science, and promoting redundant measures is an example of such noise. This would only require the authors to reanalyze their existing data (and not to collect more data) and the work is not considerable, but the findings might go against the initial predictions (and desires) of the authors which should be accepted, and which is more aligned with the scientific enterprise (e.g., focus on the evidence and not on individual preferences).

We thank the reviewer for this point, for the reference to Hodson’s interesting paper and useful recommendations, and we concur. This correlation is indeed very high (and it fails to meet even our own criteria). So we investigated the issue again and now describe it in more detail. 

We conducted a CFA comparing the single factor model with the two-factor one. Looking at the model fit, the two factor solution explains variance a lot better than the single factor model (Δχ2(1) = 804.43, p < .001). We report this in the paper. But that does not explain why the correlations between the two are so high (also in the SEM with a covariance of .89). And this is clearly a potential problem judged by Hodson’s second consideration for assessing whether a construct is really something “new” or not: “If a theoretical construct is purportedly new, independent, or distinct (theoretically), it should not correlate perfectly, or near perfectly, with another construct empirically”. But the real point is that when inspecting the model output more closely, we can see very clearly why those correlations between constructs/LV’s are so high. See the table below for the (standardized) factor loadings of the CFA where all the items of the feeling heard and intimacy scales are set to load on one single factor. The intimacy scale consists of 20 items, with factor loadings ranging from quite low (|.31|) to very high (|.91|). Among the highest loading items are the items which we could also have included in the feeling heard scale, such as “the other person was willing to listen to me” (with a loading of .91) and “the other person was open to my ideas” (.89). And clearly these are things that one might be experiencing in an intimate relationship. But crucially, the items that have greater face validity for intimacy load much less highly, such as “the other person did not want a deeper relationship with me” (-.55), “the other person seemed to care if I liked him/her” (.64) and “the other person was not attracted to me” (-.31). So what we conclude from all this is that the feeling heard scale correlates highly with the items on the intimacy scale that also measure things very similar to “feeling heard”, but the FHS does not correlate highly with the items on the intimacy scale measuring “intimacy”. Thus, feeling heard is distinct from intimacy, but the intimacy scale confounds the two and its central dimension (at least in our dataset) appears to be feeling heard. We still feel justified in concluding that this meets Hodson’s recommendations, also because of his fifth recommendation: “Statistical measurement concerns are necessary to satisfy but are not sufficient; theory must also guide decisions. [...] Construct validity is assessed by both theoretical and measurement (or statistical) means; neither is sufficient on its own, and when one is weak there is increased burden on the other to do the heavy lifting.” 

We now report this in the paper as well, see lines 757-779:

 “To check whether feeling heard and intimacy are actually measuring the same underlying construct, we compared the fit of the model where all items of the feeling heard and intimacy scales were set to load on one single factor with the model where feeling heard and intimacy were considered two separate factors. The results show that the model fit is significantly better for the two-factor solution as compared to the one-factor model (Δχ2(1) = 804.43, p < .001 ), but still the two latent factors are strongly related (covariance of .89), which suggests that the constructs might be redundant (Hodson, 2021). Inspecting the single-factor output more closely, reveals why the correlations between constructs and latent factors are so high. The intimacy scale consists of 20 items, with standardized factor loadings ranging from quite low (|.31|) to very high (|.91|). Among the highest loading items are the items which we could also have included in the feeling heard scale, such as “the other person was willing to listen to me” (with a standardized loading of .91) and “the other person was open to my ideas” (.89). And clearly these are things that one might be experiencing in an intimate relationship. But crucially, the items that have greater face validity for intimacy load much less highly, such as “the other person did not want a deeper relationship with me” (-.55), “the other person seemed to care if I liked him/her” (.64) and “the other person was not attracted to me” (-.31). In sum, the feeling heard scale correlates highly with the items on the intimacy scale that also measure experiences very similar to “feeling heard”, but the FHS does not correlate highly with the items on the intimacy scale measuring “intimacy”. This means that the feeling heard is distinct from intimacy, but the intimacy scale confounds the two and its central dimension (at least in our dataset) is feeling heard. This observation is further supported by the differing correlation patterns of feeling heard and intimacy with other measures, see the section “Feeling heard and intimacy” below.”

Item Factor loadings

Feeling heard scale: In this conversation, … 

…I felt heard by the other 0.79

…I could say what I really wanted to say 0.65

…the other was more concerned with him/herself than with what I said -0.68

…the other listened to what I said 0.81

…the other tried to put him/herself in my shoes 0.70

…the other was insensitive to my thoughts and feelings -0.68

…the other treated me with respect 0.84

…we understood each other 0.83

Intimacy scale: The other person … 

...was intensely involved in our conversation 0.61

...did not want a deeper relationship with me -0.55

...was not attracted to me -0.31

...seemed to find our conversation stimulating 0.70

...communicated coldness rather than warmth -0.80

...created a sense of distance between us -0.78

...acted bored by our conversation -0.63

...was interested in talking to me 0.85

...showed enthusiasm while talking to me 0.82

...made me feel he/she was similar to me 0.80

...tried to move the conversation to a deeper level 0.61

...acted like we were good friends 0.72

...seemed to desire further communication with me 0.81

...seemed to care if I liked him/her 0.64

...was sincere 0.83

...was interested in talking with me 0.87

...wanted me to trust him/her 0.67

...was willing to listen to me 0.91

...was open to my ideas 0.88

...was honest in communicating with me 0.80

4. There are also smaller changes, including the following:

A. perhaps report one-sample t-tests when reporting mean levels of feeling heard to indicate whether it differs from the scale middle point

We thank the reviewer for this suggestion and added these accordingly, see lines 575-577 and 722-724:

“The distribution of feeling heard in Part B was negatively skewed (skewness = -1.10; M = 3.92, SD = 0.82) and the mean level of feeling heard was significantly higher than the scale midpoint (t(213)= 16.37, p < .001, Cohen’s d = 1.12, large effect).”

“A one-sample t-test showed that the mean level of feeling heard was significantly higher than the scale midpoint (t(999) = 16.51, p < .001, Cohen’s d = .52, medium effect).”

B. in case the authors retain the additional intimacy analysis reported in p. 35, perhaps they could examine this more formally by running a moderation analysis with sample type as the moderator; this has the added benefit of avoiding splitting the sample variable.

We thank the reviewer for the suggestion. Of course those analyses could be conducted (and we did). The problem is that we then have to explain to readers all main effects, two-way interactions and the three-way interaction. The current presentation is much more elegant and direct, we feel.

C. It is unclear to me how there were “214 participants for Part B” as reported in p. 26. Please double check this. I also found the description of the conditions and two parts in p. 13 confusing, and you might want to improve it.

The 214 participants for Part B is correct. We had initially 217 participants, 3 of whom provided a clearly non-serious description of their last online conversation, which is part B. Removing these observations resulted in 214 participants for Part B. For part A, the final sample was 194 participants because there we had to exclude 23 observations – those that gave a non-serious description of a conversation in which they did (not) feel heard and those that indicated to have never experienced an online conversation in which they did (not) feel heard.

We do understand the reviewer’s confusion regarding the design description of Study 1 and clarified this on lines 286-293 (also see our response to comment 7 by Reviewer 2):

“The survey consisted of two parts. In Part A participants were either asked to think back to an online conversation in which they felt heard (N = 108) or one in which they did not feel heard (N = 86). This division was based on random allocation and done to maximize the variance in feeling heard experiences and to test the differences between (feeling heard and feeling not heard) conditions. In Part B we examined the natural distribution of feeling heard in online conversations by asking all participants to think back to their last online conversation. All participants completed both Part A and Part B but in randomized order. Eighty-nine participants began with Part A and 105 participants began with Part B.”

D. It is unclear to me the interpretation that “all items tapped into feeling heard” based on the t-test reported in p. 22

We mean that all items distinguish between feeling heard versus non-heard experiences and therefore must be measuring feeling heard. We tried to clarify this on lines 473-474:

“This suggests that all items tapped into the experience of feeling heard as they were successful in distinguishing between feeling heard versus non-heard experiences.”

E. The assertion that item 1 “loads somewhat on various factors” is misleading as Figure 1 clearly indicates it loads reliably only on two factors (respect and common group).

We agree and therefore removed this statement. 

F. I also noticed some typos/grammar issues (e.g., “participants / others that” instead of “participants / others who”; “study 1”, inconsistency in using “confirmatory factor analysis” and COVID with initial capital or not; sentence with many ‘understanding’ in line 222)

We thank the reviewer for pointing this out. We resolved these typos and grammar issues.

Reviewer #2: This study sought to develop a scale to measure feeling heard (FHS). Feeling heard is described as a complex construct of five components at two levels: at the interpersonal level, having voice; receiving attention, empathy, and respect; and at the communal level, experiencing common ground. These are described as involving me, you, and we.

Content

1. The abstract does not identify the two levels: interpersonal and collective. It seems this would be important to the reader, so state this in the abstract. Perhaps more problematic, the abstract currently seems to imply two sets of five elements (“five components at two levels”) and the two levels suggests higher-order factor structures while two conceptual levels are in fact intended. But there are actually three personal contexts: me, you, and us. So this may be better described as three interpersonal contexts: “individuals feel heard when 1) my voice receives 2) your attention, 3) empathy, and 4) respect in the context of 5) our common ground.

We thank the reviewer for these suggestions. We now explicitly mention in the Abstract and Discussion that we refer to two conceptual levels, see lines 28-30 and 584-585:

“Based on an integrative literature review, feeling heard is conceptualized as consisting of five components at two conceptual levels.”

“This scale does, however, contain elements of all five components and therefore taps into three entities (me, you, and we) at two conceptual levels (interpersonal and collective).”

Further, as indicted in our reply to comment 2B of Reviewer 1, we adapted our model description in the Introduction to de-emphasize the collective-interpersonal levels distinction in favor of focusing on the five components that concern the three interpersonal contexts mentioned by the reviewer.

Lastly, and as also indicated in our reply to comment 2A of Reviewer 1, we deliberatively chose to distinguish between interpersonal, which is about interaction partners as individuals (“me and you”), and collective, which is about the interaction partners as one irreducible entity (“us or we”). As indicated in our response to comment 2A of Reviewer 1, we now explicitly state this in the manuscript as well.

2. A key point in the discussion—lines 571 and 832—is that “feeling heard appears to be a unitary concept.” That needs to be said in the abstract as well. At present the abstract leaves the impression that feeling heard and the resulting FHS is five-dimensional.

We agree and added a short phrase stating this to the abstract, see lines 32-34:

“In two population surveys (N = 194, N = 1000), we find that feeling heard is a unitary concept and develop and validate the feeling heard scale (FHS); a concise eight-item scale with good psychometric properties.”

3. Figure 1 line 234. It seems oddly inconsistent and a bit disconcerting to insert tables into the text and place figures following the references—see note #1 under Style below.

To the best of our knowledge, this is in line with the journal’s submission requirements (https://journals.plos.org/plosone/s/submission-guidelines): “- Figure captions are inserted immediately after the first paragraph in which the figure is cited. Figure files are uploaded separately. -Tables are inserted immediately after the first paragraph in which they are cited.” But we are happy to adapt this in case we misunderstood the requirements.

4. Implicit in Study 2 is the belief that the scale will perform in substantially identical ways in the Dutch and US samples. That it does should be explicitly stated in the discussion; that is an important finding.

We thank the reviewer for this suggestion. We now include a sentence about this in the Discussion of Study 2 on lines 863-866:

“These results are important since Study 2 employed a distinct sample in a different cultural setting as compared to Study 1 and we see that the scale performed in substantially identical ways in both (Dutch and US) samples.”

5. Line 246 should conclude “can be found in the . .”

Adapted accordingly.

6. Informed consent could not be examined due to masking materials. It seems a better plan is needed to support review of this feature of the study. Briefly describe the consent procedures for each set of data.

This information will be available upon publication of the manuscript and as such (also taking readability into consideration) we decided to not include it in the text. We do understand the reviewer’s request for clarification and therefore shortly describe the procedure below. 

Concerning Study 1, the procedure was as follows: Participants read the study information within the online survey environment. The study information explained that they would be asked to describe their last online conversation and a conversation in which they did (not) feel heard and report on their experiences during these conversations. We additionally included information about the demographical questions they would be asked and our procedure for anonymizing their data. They were further informed about the average duration of the study, their compensation and reminded of the fact that their participation was entirely voluntary (and they could stop at all times). After this, participants gave their informed consent within the online survey environment. Upon completing the questionnaire, participants were fully debriefed as to the exact purposes of this study. For Study 2, the procedure was the same, except that the study information stated that participants would be asked to think back and describe one “conversation with another person or with multiple people”, instead of two online conversations.

7. The 2 x 2 design is not clear. What steps were taken to ensure that the same conversation was not reported in Part A and Part B?

We understand both reviewers’ confusion concerning the description of our design and therefore rephrased this on lines 286-293, as indicated in our response to comment 4C by Reviewer 1.

We did not take any explicit measures to prevent people from reporting the same conversation in Part A and Part B, so this could have happened. An examination of the qualitative data shows that around 6 participants repeated their (not) feeling heard description in their last conversation description, and 4 participants used their last conversation description also as the feeling (not) heard situation. This shows that same-conversation-reporting only happened in about 10 out of (194 valid observations for feeling (not) heard + 214 for last conversation) 408 descriptions. Nevertheless, the fact that some participants wrote down the same experience twice could threaten the accuracy of our assessments of natural occurrence of feeling heard experiences as well as the reliability of the scale (see the measurement invariance analysis). We tackle both of these issues in Study 2, however, by assessing both the natural occurrence of feeling heard experiences and the scale reliability in an independent sample. As indicated in response to comment 20, we now also explicitly acknowledge this in the manuscript as well.

8. Rather than state you “tried to include one negatively phrased item,” report the number of negatively-worded items in the initial and final FHS scales. The insertion of a correlation between negatively-worded items suggests there is a method effect for negatively-worded items. That might be good to include in the discussion. There is a growing body of evidence that negatively-worded items do not work similarly across some cultural settings (e.g., see Bufford & Paloutzian, 2023).

First, we rephrased the “tried to include one negatively phrased item” to “We also included four negatively phrased items.”, see line 316.

Second, considering the method effect of negatively-worded items, the 2 items’ distinctive negative phrasing might indeed explain their high correlation. We do not think there is a method effect, however, since these negative items still correlate very strongly with the other, positively-worded items as well and load on the same single feeling heard factor. We do concur with the reviewer in a more general sense in that the scale’s functioning across different cultural settings should be tested. We now include a statement about the limited demographical (also cultural) diversity of our sample in the General Discussion, see lines 1012-1014 (also see our response to comment 35):

“This is related to the relatively limited demographical diversity in our sample (often younger adults, highly educated, likely relatively affluent and mostly white).”

9. The section about descriptions and reasons—lines 342-344—seems redundant. Or perhaps clarification is needed.

We agree with the reviewer and removed this sentence. 

10. Line 347 might read better as the average of both coders or the mean score of both coders.

Adapted accordingly, see line 354.

11. Perhaps it would be worth noting in the discussion that underlying being heard are a set of shared values: respect for the person, sharing of power, a sense of justice carried out, common ground/worldview, shared being (me, you, us). And perceptions of this mutuality is essential to feeling heard.

We value this comment and think it is interesting to pursue in future research but consider it beyond the scope of the current study, as we did not assess values, and as such decided to not include this in the paper.

12. It would help to carry through with the enumeration begun in line 359. What comes second and third?

Adapted accordingly.

13. Perhaps add here that over half of the remembered Last conversations were with colleagues and family.

Added on lines 373-374:

“Indeed, over half of the remembered Last conversations were with colleagues and family.”

14. Table 3 needs a bit more detail. For percentages, it is not clear what the underlying referent is.

The percentages are fractions of the total number of described conversations in each category, so 206 for Last, 103 for Felt heard and 80 for Felt not heard conversations. We added this to the notes under Table 3, see lines 381-383.

15. Line 457 is not a complete sentence. Perhaps combine with the previous sentence.

Adapted accordingly (see lines 463-465):

“In sum, the themes and sub-themes that emerged from this content coding of the reasons’ descriptions overlap considerably with our definition (and scale) of feeling heard, except for empathy, which is not a distinct theme but more a special case of understanding.”

16. At lines 468 to 470 absolute values of t would be clearer. As stated, a t of zero is included as significant.

We changed this (lines 472-473):

“(ranging from t(124.84) = |23.12|, p < .001 to t(165.83) = |6.17|, p < .001)”

17. I suggest stating that confirmatory factor analysis rather than exploratory factor analysis was carried out because of the theoretical model for feeling heard that shaped the study.

We added the suggested sentence on lines 481-482:

“Because the scale was build upon our theoretical model of feeling heard, we used confirmatory factor analysis rather than exploratory factor analysis.”

18. Given the length and complexity of the supplementary materials, at times the reader needs more help in navigating them. A table of contents at the front of the supplementary material might work. Or refer to the page(s) on which the supplemental material is presented. A header and pagination are also essential for this material.

We thank the reviewer for this helpful suggestion. We added page numbers and a table of contents to the supplementary material.

19. Effect size data are needed for the analyses of variance at lines 545-548.

We added the Cohen’s d for both comparisons. We also included this in the matching analysis of Study 2, see lines 554-557 and 853-856:

“(F(3,190) = 5.74, p < .001; Mequal = 3.70, 95%CI = [3.51, 3.88] vs. Mother-authority = 3.26, 95%CI = [2.96, 3.56], Cohen’s d = .44, small effect, and Mself-authority = 3.30, 95%CI = [2.59, 4.01], Cohen’s d = .39, small effect).”

“(F(3,996) = 15.211, p < .001; Mequal = 3.63, 95%CI = [3.56, 3.71] vs. Mother-authority = 3.24, 95%CI = [3.11, 3.37], Cohen’s d = .41, small effect, Mself-authority = 3.25, 95%CI = [3.09, 3.61], and Mdon’t-know = 2.82, 95%CI = [2.48, 3.16] , Cohen’s d = .31, small effect).”

20. Limitations of Study 1 are not mentioned in the discussion. An important one is that the test for generality of the measure across samples (“invariance”) is quite weak as the same sample was employed: generality is only assessed across conversation settings within the same sample.

This is a valid point and related to the reviewer’s previous point about the same discussions potentially being described. We therefore now described these analysis as being a tentative indication and suggestive evidence of the replicability of the scale (see lines 566-568 and line 571):

“The single-factor model fitted the data well and the loadings were very similar to Part A. Part B also allowed a tentative indication of the measurement invariance of the scale …”

“This suggests that the scale's psychometric characteristics replicate …”

 As also indicated by the reviewer in comment 28, the confirmatory FA in Study 2 indeed provides a better assessment of the scale’s replicability and reliability, which we now explicitly acknowledge as well on lines 586-589:

“Providing initial evidence of the reliability of the scale, a good fit to Part B of the dataset was obtained. This is of course imperfect evidence since the same sample was involved. Study 2 therefore examines whether the scale is also reliable in another, unrelated dataset.”

21. The reference to procedural and distributive justice would benefit from a brief explanation—an appositive—of these concepts (justice processes and outcomes?) and their relevance to feeling heard.

We tried to clarify this in the Introduction (lines 149-152) and Discussion of Study 1 (lines 596-598): 

“Procedural justice is the fairness of the process by which a decision is made, and as such is contrasted with distributive justice which refers to the fairness of the decision itself [27,28]. Subjective procedural justice involves the experience of being treated fairly, which might be related to feeling heard.”

“That is, people can feel heard when they do not accomplish their goals (no distributive justice), and, vice versa, people can feel not heard even though they do get what they asked for (distributive justice) because of the way they are treated (procedural justice).”

22. At line 589 place convergent and divergent validity together rather than separating them with predictive validity. Placing predictive validity last would better parallel the following hypotheses as well.

We adapted the text accordingly, see lines 603-604:

“Study 2 aimed to establish the 1) reliability, 2) convergent and divergent validity, and 3) predictive validity of the feeling heard scale.”

23. The divergent validity hypothesis at line 608 is problematic. First, it purports to test the null hypothesis of no relationship. Second, divergent validity is more accurately a negative relationship rather than the absence of a relationship. Discriminant validity? Hypothesis 2B seems to best fit divergent validity as I understand it. A pertinent methodological citation(s) may be helpful as these concepts seem to be defined and used in inconsistent ways.

We agree with the reviewer that this should be discriminant validity rather than divergent validity. We replaced all mentions of divergent validity with discriminant validity.

24. The intentions for a next conversation measure is gathered as concurrent data: intentions are held in the present. Considering it a measure of predictive validity stretches the concept. 675-678

Intentions are imperfect yet often used for assessing predictive validity. We chose this measure predominantly for practical reasons: it was impossible for us to assess actual behavioral consequences over time. It would be very important for future studies to assess whether and under what circumstances these intentions translate into actual behavior. We now acknowledge this in the General Discussion on lines 947-950:

“Of course, intentions are held in the present and do not always translate into actual behavior (i.e., the intention-behavior gap). It is therefore imperative that future studies map the behavioral consequences of feeling heard experiences over time to further establish the scale’s predictive validity.”

25. A period seems to be in the wrong place at line 686. Actually several; check punctuation in this entire paragraph.

Adapted accordingly.

26. The data at 679-689 lack needed context. What are these relationships?

Here relationship refers to the relationship between the participant and the interaction partner(s) in the conversation the participant remembered and described. We tried to clarify this on lines 695-697:

“Within these reactions, we distinguished between positive approach/avoidance: behaviors that steer towards maintenance of the relationship between the participant and their interaction partner(s)…”

27. It seems unclear how the variables following 693 fit the concept of divergent validity. Perhaps a technical citation is needed here as it appears the literature seems to conflate divergent validity and discriminant validity. See #23 above.

See our reply to comment 23. We adapted all references to discriminant validity.

28. The CFA for Study 2 is much more valuable than that for Study 1 as it employs a distinct sample in a distinct cultural setting. This should be mentioned in the discussion of Study 2 or General Discussion.

We agree with the reviewer and, as also indicated in response to comment 4, added a sentence stating this in the Discussion of Study 2. As indicated in our response to comment 20, we also refer to this in the Discussion of Study 1.

29. Table 5 would be more effective if it were constructed to parallel the hypotheses 2A and 2B. The present format makes the reader work to follow the expected direction of relationships and minus signs are scattered rather than clustered together.

We implemented the reviewer’s suggestion, see Table 5.

30. Line 735 claims that “feeling heard differs from intimacy”, but it appears this claim was not statistically tested—are these correlations significantly different? Perhaps simply report they show somewhat different patterns of correlation with other measures.

Adapted, see lines 777-779:

“This observation is further supported by the differing correlation patterns of feeling heard and intimacy with other measures, see the section “Feeling heard and intimacy” below.”

31. While the logic of regression is predictive, data were gathered at the same time and is somewhat misreported at line 745. It would be more accurate to say “The less people felt heard, the more likely they they were to say that they were disposed to negatively avoid a next conversation. . . “

We adopted the reviewer’s suggestion (lines 791-792):

“The less people felt heard, the more likely they were to say that they were disposed to negatively avoid a next conversation with the same person(s)…”

32. Taken as a whole the regressions establish the FHS as a predictor of engaging/avoiding conversation and item 1 is a good short version. And yes, feeling heard is distinct from the related constructs. 778-779

We agree.

33. The claim that “not feeling heard appears to be somewhat less common than feeling heard” is not tested in the study (line 845). It seems all you could claim here is that participants were more likely to report an episode of feeling heard. Might that be due to the relative rarity of feeling heard? Could is be influenced by the initial instructions?

We agree with the reviewer and adapted the phrasing on lines 894-898 accordingly:

“Across different contexts and relationships, we find that when people reflect on a significant social interaction they recently had, they are more likely to report that they did feel heard than did not feel heard. Although we can not say this with certainty, this suggests that across many individuals, there might be a base rate for people to feel heard in most of their conversations.”

34. At line 891-896 the supporting data are not clear. Does the conversational intimacy comment pertain to a finding I overlooked—or to another study which should be cited here?

This refers to the data concerning the differences between feeling heard and intimacy in predicting negative avoidance intentions within 2 different subsamples: conversations in which participants felt positive about their relationship with their interaction partner(s) and conversations in which this was not the case, see lines 833-835.

35. Characteristics of the sample seem additional limitations—young adults, more educated, likely more affluent, likely mostly white—I did not see any ethnic/racial description of the samples.

As stated in the manuscript, we recruited a sample that is representative for the Dutch population, so the limited demographic diversity mostly applies to Study 2. For ethical and data minimalization reasons, we did not assess ethnicity or race, but this might indeed be rather uniformly white. Race might indeed affect participants’ understanding of feeling heard. As indicated in our reply to comment 8, we included a remark about this in the limitations concerning demographic diversity (lines 1012-1014).

36. The supplementary material is a bit chaotic. Adding pagination, a table of contents, and page breaks that fit the content would help to make this material (all 51 pages) accessible.

As indicated under comment 18, we made these adaptations.

B. Format/Style

1. In APA style, tables and figures can be embedded in the text or aggregated after the reference list. Here neither of these options is employed; rather they are split in both places.

See our response to comment 3 above.

2. See line 288—in APA style numbers are spelled out to begin a sentence.

Adapted. 

3. Embedded tables should be preceded and followed by an extra double line-space (e.g., 354-358).

We applied this to all Tables in the manuscript.

4. Single-spaced tables are acceptable in APA style, but an additional line space between sections would make the material more readable (e.g., Table 2, 356ff).

We applied this to all tables in the manuscript. 

5. Correlational tables are generally constructed with the data on the lower left part of the table rather than the upper right half.

We adapted the correlation table in the manuscript (Table 5) according to the reviewer’s comment. 

6. Figures require a descriptive title and sufficient information in the notes to explain them independent of the text. The header should remain constant except for the page #. Figure titles are placed below the figures. For example, lines 531-536 should be included on the page with the figure. In this manuscript they appear to be inserted into the text without the needed extra spacing to distinguish them from the text.

See our response to comment 3 above.

7. Table S-2 would be more accessible if t and its df were placed in adjacent columns or t and df were listed sequentially like M, (SD). Several of the supplementary tables would also benefit by putting additional material in the same or an adjoining column rather than between rows.

We thank the reviewer for pointing this out and adapted all Tables where space permitted.

---

## [Decision Letter · Decision Letter 1]

2 Oct 2023

Feeling Heard: Operationalizing a key concept for social relations

PONE-D-23-13419R1

Dear Dr. Roos,

We’re pleased to inform you that your manuscript has been judged scientifically suitable for publication and will be formally accepted for publication once it meets all outstanding technical requirements.

Kind regards,

Ramona Bongelli, Ph.D.

Academic Editor

PLOS ONE

Additional Editor Comments (optional):

Reviewers' comments:

Reviewer's Responses to Questions

**Comments to the Author**

1. If the authors have adequately addressed your comments raised in a previous round of review and you feel that this manuscript is now acceptable for publication, you may indicate that here to bypass the “Comments to the Author” section, enter your conflict of interest statement in the “Confidential to Editor” section, and submit your "Accept" recommendation.

Reviewer #1: All comments have been addressed

2. Is the manuscript technically sound, and do the data support the conclusions?

Reviewer #1: Yes

3. Has the statistical analysis been performed appropriately and rigorously? 

Reviewer #1: Yes

4. Have the authors made all data underlying the findings in their manuscript fully available?

Reviewer #1: Yes

5. Is the manuscript presented in an intelligible fashion and written in standard English?

Reviewer #1: Yes

6. Review Comments to the Author

Reviewer #1: It is clear the authors considered all the comments made by the reviewers and have submitted a much stronger manuscript.

7. PLOS authors have the option to publish the peer review history of their article (what does this mean?). If published, this will include your full peer review and any attached files.

Reviewer #1: No

---

## [Editor Report · Acceptance letter]

6 Oct 2023

PONE-D-23-13419R1 

Feeling Heard: Operationalizing a key concept for social relations 

Dear Dr. Roos:

I'm pleased to inform you that your manuscript has been deemed suitable for publication in PLOS ONE. Congratulations! Your manuscript is now with our production department. 

Kind regards, 

on behalf of

Professor Ramona Bongelli 

Academic Editor

PLOS ONE